# Rethinking Regularization in Federated Learning: An Initialization Perspective

## Abstract

In federated learning, numerous regularization methods have been introduced to alleviate local drift caused by data heterogeneity. While all share the goal of reducing client drift, their effects on client gradients and the resulting features learned by local models differ. Our comparative analysis shows that among the tested regularization methods, FedDyn is the most effective, achieving superior accuracy-to-round while simultaneously reducing inter client gradient divergence and preserving global model features during local training. Nevertheless, regularization methods, including FedDyn, are only approximations of an ideal scheme that would completely remove local drift and guarantee convergence to the global stationary point. In practice, deviations from this ideal give rise to side effects and, together with the additional computational and communication costs, limit their practicality. Since the performance differences among federated learning algorithms diminish once models are well-initialized, it is more efficient to restrict regularization to the pre-training phase, where its benefits outweigh these drawbacks. Our study of pre-training strategies for FedAvg demonstrates that FedDyn provides the most effective initialization, a property tied to its convergence behavior near the global stationary point. Extensive experiments across both cross-silo and cross-device settings confirm that applying FedDyn solely for pre-training yields faster convergence and reduced overhead compared to maintaining regularization throughout the entire training process.

## 1 Introduction

Federated learning preserves client privacy by keeping data localized on edge devices, allowing each client to update models using only its own data. Directly transmitting updates after every gradient step would incur prohibitive communication costs. To address this, FedAvg (McMahan et al., 2017) was proposed, where clients perform multiple local steps before sending updates to the server. However, these multiple steps amplify the adverse effects of data heterogeneity, which arises from the inherently distinct data distributions across clients and leads to inconsistencies between local and global objectives (Li et al., 2020). To mitigate this issue, several methods have been introduced, with regularization-based approaches such as SCAFFOLD (Karimireddy et al., 2020) and FedDyn (Acar et al., 2021) among the most prominent. By introducing regularization during local training, these methods reduce the mismatch between local and global objectives. In this work, we aim to analyze the underlying mechanisms of regularization from a new perspective, asking the following key questions.

The first question we address is: *How does regularization influence client gradients?* We analyzed this from two perspectives: (i) differences in gradients across clients and (ii) changes in gradients within a single client across rounds. To quantify inter-client differences, we measured gradient diversity (Yin et al., 2018); to capture intra-client changes, we computed the cosine similarity of gradients across rounds. An effective regularization method should reduce gradient diversity by better aligning the gradients of clients that have non-IID data. At the same time, it should yield low intra-client cosine similarity, since high similarity suggests that a client repeatedly optimizes only its local objective without regard to the global objective. In the case of plain FedAvg without regularization, clients focus solely on minimizing their local objectives, resulting in updates that remain almost identical in direction across rounds but diverge significantly across clients. This growing gradient diversity ultimately slows down global convergence. Based on these measures, we found

that FedDyn exhibits the most desirable regularization behavior. In FedDyn, clients update their models in directions that are nearly orthogonal to their previous gradients, thereby accounting not only for their local objectives but also for the global objective. This results in lower inter-client gradient diversity and, consequently, superior accuracy-to-round performance. In contrast to methods that mitigate client drift only indirectly, FedDyn directly enforces alignment between local and global objectives, guaranteeing that the solutions of individual clients and the global model converge to the same point in the limit. This finding is consistent with benchmark results in (Baumgart et al., 2024), where FedDyn consistently outperforms other federated learning methods in terms of accuracy-to-round.

The second question we address is: *How does regularization influence the features learned by local and global models?* To identify which features a model has captured, we employed the interaction tensor (Jiang et al., 2024), which enables direct comparison of the information encoded by different models. Specifically, we computed the interaction tensor among the current global model, the local models, and the subsequent global model. Our analysis shows that, without regularization, each client primarily learns features tailored to its own data distribution, whereas regularization encourages local models to acquire features that better align with the features of the global model.

The final question we address is: *What are the drawbacks of regularization, and how can they be mitigated?* An ideal regularization method would completely eliminate local drift and ensure that updates are equivalent to those guided by the gradient of the global objective function. SCAFFOLD and FedDyn employ client-side control variates to approximate and remove local drift, while using server-side control variates to approximate the gradient of the global objective, thereby producing updates that closely mimic those of the global gradient. Naturally, a gap remains between existing regularization methods and the ideal scheme, and even FedDyn, which comes closest to ideal regularization, exhibits side effects as a result. Specifically, in FedDyn, the server control variate does not accurately approximate the gradient of the global objective function, and updates guided by it can negatively impact the features learned by the global model. In the early stages of training, the benefits of regularization outweigh its side effects, but as training progresses these advantages diminish, and when accounting for the additional computational cost, the gains become marginal. Federated learning is less sensitive to data heterogeneity when initialized with well-trained weights, such as those obtained from public datasets (Nguyen et al., 2023; Chen et al., 2023). This suggests that regularization is unnecessary in the later phases of training. To address this, we propose a two-stage approach: applying regularization only during initialization, followed by fine-tuning with FedAvg. We show that using regularization solely for initialization mitigates the aforementioned drawbacks and achieves results comparable to initialization from public datasets, which may not always be available in practice. The contributions of this work are summarized as follows:

- We conduct a comparative analysis of various regularization methods not only in terms of accuracy-to-round but also from gradient, feature learning, and initialization perspectives, providing a deeper understanding of how regularization affects data heterogeneity.

- We analyze the gap between ideal regularization and its closest practical counterpart, FedDyn, identify the side effects that arise from this gap, and validate these findings through experiments.

- We propose a two-stage training strategy that employs regularization only for pre-training, motivated by the observation that the impact of data heterogeneity diminishes once weights are well-initialized and that regularization introduces additional computational cost.

## 2    RELATED WORK

### 2.1    HETEROGENEITY IN FEDEATED LEARNING

Data heterogeneity introduces inconsistencies between the local objective of each client and the global objective. To address this issue, regularization-based methods such as SCAFFOLD (Karimireddy et al., 2020), FedDyn (Acar et al., 2021), FedNTD (Lee et al., 2022), and MOON (Li et al., 2021) have been proposed. These approaches encourage clients to move beyond optimizing solely for their local objectives and instead contribute to the convergence of the global model toward a global optimum. However, compared to FedAvg (McMahan et al., 2017), such methods incur additional computational overhead, which can be burdensome for resource-constrained clients. Federated learning is commonly categorized into two scenarios: Cross-Silo and Cross-Device (Kairouz

et al., 2019). In Cross-Silo FL, a relatively small number of clients participate, each often holding a substantial amount of data, for example, hospitals collaboratively training on medical data (Rieke et al., 2020). In contrast, Cross-Device FL typically involves millions or even billions of clients (Niu et al., 2020), each contributing sporadically with a small amount of data, such as smartphones participating in keyboard prediction tasks (Hard et al., 2018). This paper empirically investigates the role and impact of regularization in both Cross-Silo and Cross-Device settings.

## 2.2 PRE-TRAINING FOR FEDERATED LEARNING

In standard federated learning, the global model is typically initialized with random weights. Recent studies have examined the role of pre-training for initialization (Nguyen et al., 2023; Chen et al., 2023), collectively highlighting that initialization is critical in federated learning. In particular, they suggest that leveraging pre-trained weights, rather than random initialization, can help mitigate the challenges posed by data heterogeneity. However, pre-training on large publicly available datasets may not always be feasible. To address this limitation, prior work has explored the use of synthetic data, such as fractal datasets, for pre-training when public data is unavailable (Chen et al., 2023). Nevertheless, such synthetic pre-training remains less effective than using large real-world datasets. In this work, we propose a pre-training methodology for federated learning that does not rely on external public datasets, but instead leverages only the distributed data available across clients.

## 3 RETHINKING REGULARIZATION IN FEDERATED LEARNING

### 3.1 NOTATIONS

We consider a federated learning scenario with $N$ clients. In each round $t$, a random subset $P_t$ participates. Each client $k \in P_t$ receives the current global model $\theta^t$ from the server, performs $E$ local epochs on its objective $L_k$, and returns the updated model $\theta_k^{t+1}$. The goal of federated learning is to minimize the global objective $L(\theta) \triangleq \frac{1}{N} \sum_{k=1}^{N} L_k(\theta)$, where the *pseudo-gradient* is defined as $g_k^t \triangleq \theta_k^{t+1} - \theta^t$, i.e., the update contributed by client $k$ in round $t$. To mitigate local drift, control variates are often introduced: a client control variate $h_k^t \approx \nabla L_k(\theta_k^t)$ that approximates the local gradient, and a server control variate $h^t \approx \nabla L(\theta^t)$ that approximates the global gradient.

### 3.2 GRADIENT PERSPECTIVE

In this work, we aim to provide direct evidence that FedAvg's convergence is hindered by local drift and to examine whether similar issues arise in other methods. To this end, we measured (i) the diversity of pseudo-gradients across clients and (ii) the cosine similarity of pseudo-gradients within each client across rounds. Gradient diversity, originally introduced in (Yin et al., 2018), quantifies how different the individual gradients of local objectives are from each other, and is defined as

$$\text{Gradient Diversity} \triangleq \frac{\sum_{k=1}^{n} \|\nabla L_k(\theta)\|_2^2}{\left\| \sum_{k=1}^{n} \nabla L_k(\theta) \right\|_2^2}. \tag{1}$$

While gradient diversity captures the diversity of pseudo-gradients across clients, cosine similarity reflects their similarity across rounds within a single client. Specifically, we measured the cosine similarity between a client's pseudo-gradient from the last round and those from its previous rounds. Experimental results obtained with ResNet20 (He et al., 2016) on CIFAR-100 are shown in Figures 1b and 1c. We also compared the accuracy of different regularization methods across communication rounds in a setting where 10% of 100 clients participated in each round. To reduce randomness, pseudo-gradient diversity and cosine similarity were computed under full client participation, whereas accuracy-to-round was evaluated under partial participation to emphasize practical relevance. As discussed later in Section 3.4, where we analyze the differences among various regularization methods used for FedAvg initialization, we switched to FedAvg after a certain round. The vertical gray line on the x-axis denotes this switching point. In this section, we analyze the training dynamics of the regularization methods prior to this point.

In Figure 1a, FedDyn and SCAFFOLD achieve the fastest convergence, while FedNTD and MOON converge at rates similar to FedAvg. This behavior can be explained by pseudo-gradient diversity

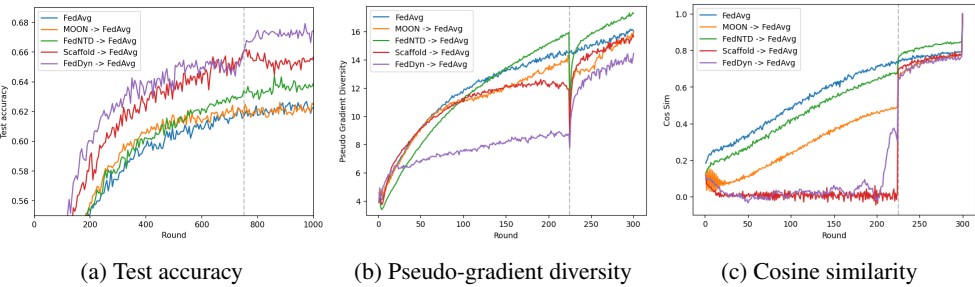

(a) Test accuracy       (b) Pseudo-gradient diversity       (c) Cosine similarity

Figure 1: Comparison of regularization-based methods in terms of (a) test accuracy, (b) pseudo-gradient diversity, and (c) cosine similarity.

and cosine similarity. As shown in Figures 1b and 1c, FedAvg exhibits the highest pseudo-gradient diversity and cosine similarity in the early rounds. High cosine similarity indicates that a client's pseudo-gradients change little across rounds, even as the global model evolves, while high gradient diversity implies that client updates cancel each other out, slowing down global progress. By contrast, FedDyn and SCAFFOLD encourage clients to update in directions nearly orthogonal to their previous updates, yielding lower pseudo-gradient diversity and faster convergence. Because Fed-NTD and MOON regularize client updates only indirectly (via knowledge distillation or contrastive learning), their pseudo-gradient diversity and cosine similarity follow trends similar to FedAvg, leading to slower convergence compared to direct methods.

Although FedDyn and SCAFFOLD show similar cosine similarity, FedDyn achieves lower pseudo-gradient diversity and slightly faster convergence. Both employ control variates to reduce client drift, but they differ in communication cost: SCAFFOLD requires exchanging both the model and the server's control variate with clients. In summary, FedDyn most effectively aligns local and global objectives, achieving the fastest convergence and strongest gradient alignment. In contrast, the indirect regularization terms in MOON and FedNTD fail to provide sufficient correction, causing clients to repeat similar updates and resulting in slow convergence. This finding is consistent with benchmark results in (Baumgart et al., 2024), where FedDyn consistently outperforms other federated learning methods, including SCAFFOLD, in terms of accuracy-to-round when computational overhead is not taken into account. Unless otherwise specified, we adopt FedDyn as the default regularization method.

### 3.3 FEATURE LEARNING PERSPECTIVE

We further analyzed how the gradient changes discussed in Section 3.2 influence the features learned by the model. To identify which features each model captures, we computed the interaction tensor (Jiang et al., 2024) $\Omega \in {0,1}^{M \times N \times F}$ among the current global model at round $t$, the locally trained models, and the subsequent global model. Here, the first axis corresponds to $M = |P_t| + 2$ models, the second axis to $N$ test data points, and the third axis to $F$ feature clusters. An entry $\Omega_{mnf} = 1$ indicates that the $m$-th model has learned the $f$-th feature and that the $n$-th test data point contains this feature. We define the top-$K$ principal components of the penultimate layer's output $\Phi \in \mathbb{R}^{N \times d}$ as features, where $d$ is the input dimension of the final fully connected layer. Each representation is projected into $\mathbb{R}^K$. Using singular value decomposition (SVD), $\Phi = U\Sigma V^T$, we select $K$ columns of $V$ as principal components. The projection of a test data point $x$ in $\mathbb{R}^K$ is denoted $v(x)$, and the $k$-th entry of $v(x)$ for the $m$-th model is written as $v_{m,k}(x)$. From the total of $MK$ features, we perform greedy clustering on features with high correlation. For any pair of models $(w_i, w_j)$ and their respective features $(a, b)$, the correlation $\rho_{(i,j),(a,b)}$ is computed as

$$\rho_{(i,j),(a,b)} = \mathbb{E}_{(x,y)\sim\mathbb{D}}\big[(v_{i,a}(x) - \mu_{i,a})(v_{j,b}(x) - \mu_{j,b})\big](\sigma_{i,a}\sigma_{j,b})^{-1}, \qquad (2)$$

where $\mu_{i,a}$ and $\sigma_{i,a}$ are the mean and standard deviation of $v_{i,a}$. When $|\rho_{(i,j),(a,b)}|$ exceeds a threshold $\gamma_{\text{corr}} \in (0,1)$, the $a$-th feature of the $i$-th model and the $b$-th feature of the $j$-th model are assigned to the same feature cluster. To identify which data points contain a given feature, we normalize $v_{m,k}$ by its $\ell_\infty$-norm. If the $k$-th entry in the $n$-th data point exceeds a threshold $\gamma_{\text{data}} \in (0,1)$, then the $n$-th data point is considered to contain the $k$-th feature of the $m$-th model. Finally, for each model

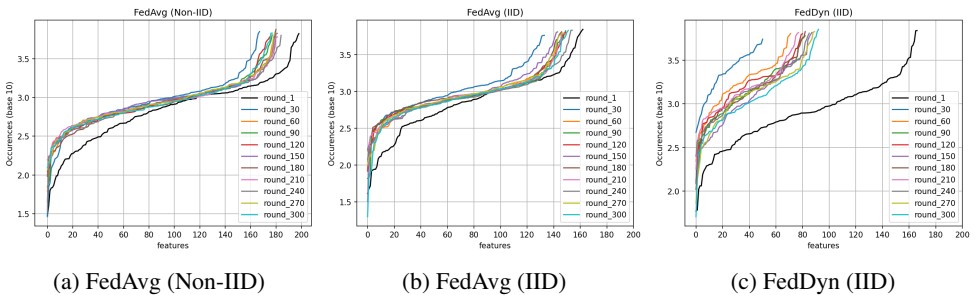

(a) FedAvg (Non-IID)       (b) FedAvg (IID)       (c) FedDyn (IID)

Figure 2: Feature frequency over training rounds

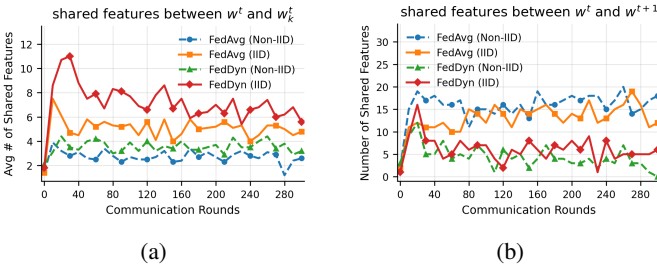

(a)                    (b)

Figure 3: (a) Average number of features in local models that also appear in the current global model. (b) Number of features shared between the current and the next global model.

$m$ and feature $k$, if there exists a data point $n$ containing the feature and this feature belongs to cluster $f$, then $\Omega_{mnf} = 1$.

We analyzed how the features learned by each model differ depending on the client's data distribution using the interaction tensor, with $d = 64$ (input dimension of the final fully connected layer) and $K = 20$ principal components. Feature occurrence frequency was measured during global training with FedAvg across 10 clients using ResNet-20 on CIFAR-100. In the Non-IID setting (Figure 2a), each model tends to learn distinct features, resulting in a larger number of feature clusters (170–180 features) throughout training. In contrast, in the IID setting (Figure 2b), the learned features exhibit stronger correlations and form fewer clusters (140–150 features).

We also investigated the effect of regularization. As shown in Figure 2c, FedDyn promotes tighter clustering of features, reducing the number of clusters to 80–90. Furthermore, we examined feature preservation across training by measuring (i) how well local models retain features of the current global model after local training, and (ii) how many of these features persist in the next global model after aggregation. In Figure 3a, local models in the IID setting largely preserve features from the current global model, whereas in the Non-IID setting they tend to learn new features not present previously. FedDyn's local models, however, effectively retain the feature representations of the global model. Finally, in Figure 3b, we observe that although FedAvg's local models diverge from the current global model, the next aggregated global model still exhibits feature representations similar to the current one. By contrast, FedDyn's global model shows greater round-to-round changes in feature representations, despite its local models preserving many of the global model's existing features.

## 3.4 REGULARIZATION FOR INITIALIZATION

Regularization reduces gradient diversity and thereby accelerates convergence. However, this effect relies on updates guided by the server control variate, which approximates the global gradient. In FedDyn, local updates are regularized to suppress drift, while the server compensates for accumulated client drifts using $h^t$. Yet, as shown in Figure 6d, these client drifts can exhibit high diversity, making the server-side compensation imprecise. FedDyn converges to a global stationary point only in the asymptotic regime (i.e., as $t \to \infty$), under the assumption that each local model converges to $\theta^\infty$. In practice, local models do not fully converge, so the condition

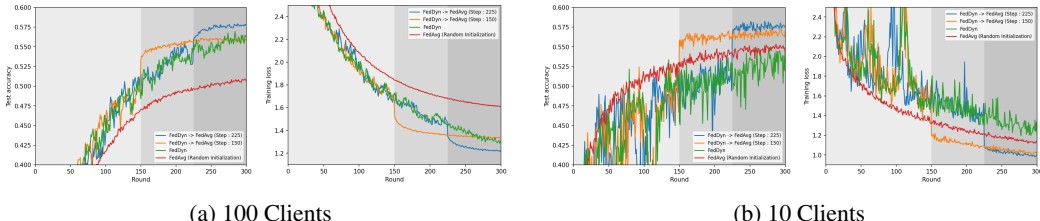

(a) 100 Clients

(b) 10 Clients

Figure 4: Test accuracy and train loss of FedAvg, FedDyn, and FedDyn → FedAvg on CIFAR-100 dataset. Train loss refers to the training loss of the global model for the entire training data.

$\sum_k \nabla L_k(\theta_k^t) \approx \sum_k \nabla L_k(\theta^\infty) \to 0$ holds only approximately. This imprecision also manifests in feature learning. As shown in Figure 3b, regularization helps local models retain features of the global model, but the aggregated update offsets this benefit, so that the next global model shares relatively few features with the current one. In the Section A.1, we provide empirical evidence that the aggregated local model $\gamma^t = \frac{1}{P} \sum_{k \in P_t} \theta_k^t$ achieves better performance than the global model $\theta^t$ obtained by a single-step update with $h^t$, demonstrating that $h^t$ does not accurately approximate the global gradient.

**Formal Switching Criterion.** Motivated by this observation, we propose switching from FedDyn to FedAvg. We also provide a formal condition for identifying when continued use of FedDyn ceases to be effective and when it is preferable to switch to FedAvg, under the assumption of convex and $L$-smooth loss functions. The key intuition is that although Theorem 2 of (Acar et al., 2021) establishes the convergence of $\gamma^t = \frac{1}{P} \sum_{k \in P_t} \theta_k^t$, the imprecise update using the server control variate $h^t$ may hinder the convergence of $\theta^t$, the actual global model distributed to clients. Therefore, if the cost introduced by $h^t$ outweighs the convergence gain from regularization, it becomes undesirable to maintain regularization. We emphasize that this switching rule is conservative, since it does not take into account any potential gain after switching to FedAvg. Formally, building on Lemma 1 of (Acar et al., 2021), we obtain the following proposition (derivation deferred to Appendix A.4). Let

$$C_t = \frac{1}{m} \sum_{k \in [m]} E\big[\|\nabla L_k(\theta_k^t) - \nabla L_k(\theta^*)\|^2\big], \tag{3}$$

which measures how well client gradients approximate the local gradients at the optimal weights. Then the averaged suboptimality after $T$ rounds can be upper-bounded as

$$D_T = \frac{1}{T}\left(\zeta + \eta \sum_{t=1}^{T} C_{t-1}\right), \qquad \zeta = \left(1 + \frac{L}{\alpha}\right)\frac{\Phi_0}{\kappa_0}, \quad \eta = \frac{1}{2\alpha} + \frac{L}{2\alpha^2}. \tag{4}$$

**Proposition 1** (Switching Point for FedDyn). *A condition for the round $T$ at which FedDyn ceases to be effective is given by*

$$D_T \le D_{T+1} \iff T\eta\left(C_T - \frac{1}{T}\sum_{t=1}^{T} C_{t-1}\right) \ge \zeta, \tag{5}$$

*where $\frac{1}{T}\sum_{t=1}^{T} E[\ell(\theta^{t-1}) - \ell(\theta_*)] \le D_T$, $\zeta$ is a positive constant.*

Thus, when the model has not sufficiently converged and $C_t$ remains large relative to its historical average, an earlier switch should be performed. As shown in Section 3.2, local gradients remain nearly aligned regardless of changes in the global model. Considering that FedDyn imposes regularization along this direction, the condition that $C_T$ exceeds its historical average can hold.

We present experimental evidence that FedDyn is not always effective as a standalone method. While it reduces client drift, its main utility lies in providing a strong initialization for FedAvg. As shown in Figure 4, we compare FedAvg, FedDyn, and FedDyn→FedAvg on the CIFAR-100 dataset using a ResNet20 (He et al., 2016) model, with data distributed non-IID across either 100 or 10 clients under full participation. The results show that applying regularization throughout training is less effective than using it only for initialization. In Figure 4a, FedDyn converges faster and

Table 1: Test accuracy of FedAvg, FedDyn, and FedDyn→FedAvg across various federated learning settings

| Data | Dataset | Cross-Device | | | Cross-Silo | | |
|------|---------|--------|--------|------------|--------|--------|------------|
| | | FedAvg | FedDyn | FedDyn→FedAvg | FedAvg | FedDyn | FedDyn→FedAvg |
| IID | CIFAR10 | $86.35_{\pm 0.12}$ | $88.97_{\pm 0.15}$ | $\mathbf{89.11_{\pm 0.16}}$ | $86.63_{\pm 0.06}$ | $88.25_{\pm 0.17}$ | $\mathbf{89.91_{\pm 0.13}}$ |
| | CIFAR100 | $62.58_{\pm 0.22}$ | $65.71_{\pm 0.20}$ | $\mathbf{67.18_{\pm 0.49}}$ | $66.30_{\pm 0.14}$ | $66.03_{\pm 0.27}$ | $\mathbf{68.99_{\pm 0.23}}$ |
| | Tiny-ImageNet | $52.39_{\pm 0.38}$ | $56.04_{\pm 0.15}$ | $\mathbf{57.85_{\pm 0.25}}$ | $57.30_{\pm 0.09}$ | $55.96_{\pm 0.21}$ | $\mathbf{59.51_{\pm 0.22}}$ |
| NonIID | CIFAR10 | $73.36_{\pm 0.38}$ | $\mathbf{81.83_{\pm 0.39}}$ | $81.43_{\pm 0.44}$ | $62.37_{\pm 0.50}$ | $75.39_{\pm 0.35}$ | $\mathbf{78.29_{\pm 0.24}}$ |
| | CIFAR100 | $53.51_{\pm 0.28}$ | $54.25_{\pm 0.69}$ | $\mathbf{56.48_{\pm 0.24}}$ | $54.21_{\pm 0.14}$ | $53.98_{\pm 0.74}$ | $\mathbf{57.77_{\pm 0.53}}$ |
| | Tiny-ImageNet | $50.28_{\pm 0.75}$ | $54.19_{\pm 0.36}$ | $\mathbf{54.87_{\pm 0.41}}$ | $52.68_{\pm 0.21}$ | $51.97_{\pm 0.38}$ | $\mathbf{53.55_{\pm 0.22}}$ |

reaches higher accuracy than FedAvg, whereas in Figure 4b this advantage disappears, suggesting that the benefit of regularization depends on the number of clients and the size of local datasets. Nonetheless, across both settings, weights trained with FedDyn provide excellent initialization for FedAvg. Although FedDyn achieves the best accuracy-to-round, it is also the most computationally expensive (Baumgart et al., 2024). Using it only for pre-training therefore alleviates this burden. Longer pre-training improves final performance but also increases overhead, making the number of pre-training rounds a trade-off between cost and generalization. Since FedAvg converges rapidly once initialized, it is generally advantageous to retain regularization for as long as possible. In our experiments, we applied it for 75% of the total rounds, with an ablation study of the switching point provided in Section A.2.

We also examined whether other regularization methods could serve as good initialization strategies for FedAvg, but as shown in Figure 1a, this was not the case. As reported in (Nguyen et al., 2023), pre-trained weights exhibit lower pseudo gradient diversity compared to random initialization. Similarly, as shown in Figure 1b, only the FedDyn → FedAvg strategy demonstrates this property, which we attribute to FedDyn approximately converging to the global stationary point.

## 4 EXPERIMENT

### 4.1 EXPERIMENT SETTING

We evaluated our approach on image classification tasks using multiple datasets and models: VGG11 (Simonyan & Zisserman, 2014) for CIFAR-10, ResNet20 for CIFAR-100, and ResNet18 (He et al., 2016) for Tiny-ImageNet. The corresponding hyperparameters are listed in Section A.9. After pre-training, we always fine-tune with FedAvg. The ablation study on methods other than FedAvg is provided in Section A.3. The procedure for applying dynamic regularization (Acar et al., 2021) as initialization for FedAvg is summarized in Algorithm 1: training follows FedDyn until a specified pre-training step, after which it switches to FedAvg. To model non-IID data, we used Dirichlet-based partitioning, where the data of class $c$ assigned to client $k$ follows $p_c \sim \text{Dir}_k(\beta)$. We set $\beta = 0.5$ for CIFAR-10 and $\beta = 0.1$ for CIFAR-100 and Tiny-ImageNet. All experiments were repeated three times, and we report mean and standard deviation.

### 4.2 CROSS-DEVICE AND CROSS-SILO

Table 1 shows that initializing FedAvg with weights pre-trained by FedDyn is consistently effective across both system settings (Cross-Device and Cross-Silo) and data distributions (IID and non-IID). In the Cross-Device setting, FedDyn exhibits superior convergence and higher test accuracy compared to FedAvg, yet applying regularization throughout all rounds proves less beneficial than switching to FedAvg after a certain point. In the Cross-Silo setting, FedDyn does not clearly outperform FedAvg, but surprisingly, FedDyn pre-training still provides a strong initialization that improves FedAvg performance.

These results hold regardless of data distribution. FedAvg initialized with regularization achieves substantially higher accuracy than starting from random initialization, in both IID and non-IID cases. This suggests that the primary challenge FedAvg faces under data heterogeneity stems from its initialization. Our finding resonates with recent work highlighting the importance of pre-training

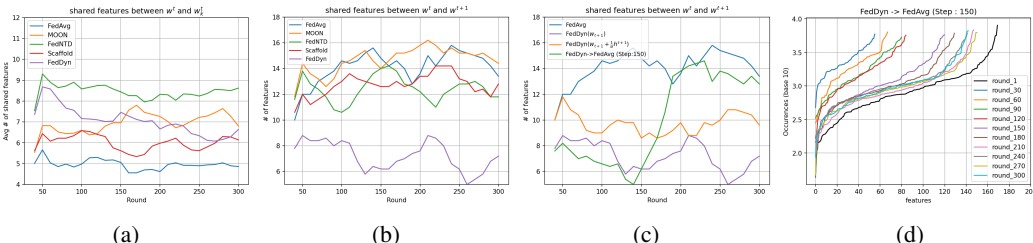

(a)  (b)  (c)  (d)

Figure 5: (a) represents the number of features shared between the current global model and local models, while (b),(c) represents the number of features shared between the current global model and the next global model. (d) represents the number of feature clusters over training rounds.

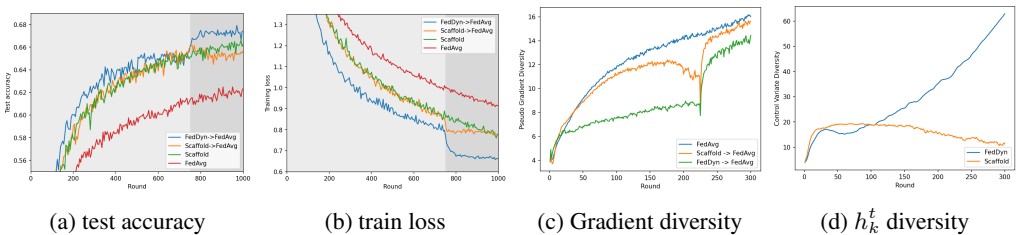

(a) test accuracy  (b) train loss  (c) Gradient diversity  (d) $h_k^t$ diversity

Figure 6: Comparison of Scaffold and FedDyn from the initialization perspectives on CIFAR100 dataset

in federated learning (Nguyen et al., 2023; Chen et al., 2023), while differing in methodology: our approach leverages model weights pre-trained within the federated environment itself, rather than those obtained in a centralized setting with external datasets.

## 4.3 REGULARIZATION METHOD

In Section 3.2, we described dynamic regularization as the most ideal form of regularization. We next examined this claim from the perspective of feature learning. In Figure 5a, we compare how well local models trained with different methods preserve the features of the current global model. FedNTD (Lee et al., 2022), which uses knowledge distillation, retains global features most effectively, while FedDyn also preserves them well, reaffirming the desirable properties of dynamic regularization.

In Figure 5b, we analyze how much the next global model retains the features of the current global model. Here, all methods except FedDyn behave similarly to FedAvg. To isolate the effect of the server control variate, we measured feature retention before applying the server update. As shown in Figure 5c, the reduced retention in FedDyn is caused by the compensation step using $h^{t+1}$. When fine-tuning with FedAvg after the pre-training step, the next global model not only preserves the current global features but also, as shown in Figure 5d, expands the total number of feature clusters.

Next, we compared FedDyn and Scaffold as initialization methods, given their similarities (Figure 6). Unlike FedDyn, Scaffold does not provide a good initialization point: FedAvg pre-trained with Scaffold performs worse than Scaffold itself. Still, for both methods, switching to FedAvg typically produces a sharp drop in training loss. The reason only FedDyn serves as a good initialization point for FedAvg is that FedDyn converges toward an approximated global stationary point, i.e., $\sum_k \nabla L_k(\theta_k^t) \to 0$. This property ensures that the weights obtained after FedDyn pre-training provide a stable and well-aligned starting point for subsequent FedAvg fine-tuning. The differences also appear in the diversity of pseudo gradients and control variates (Figures 6c and 6d). While both reduce pseudo gradient diversity compared to FedAvg, FedDyn achieves much lower diversity, and Scaffold's control variate diversity does not increase noticeably.

Beyond these cases, federated learning includes many optimizers such as MOON (Li et al., 2021), FedNTD (Lee et al., 2022), FedNOVA (Wang et al., 2020b), FedMA (Wang et al., 2020a), FedAVGM (Hsu et al., 2019), and FedADAM (Reddi et al., 2021). Thus, numerous combinations of

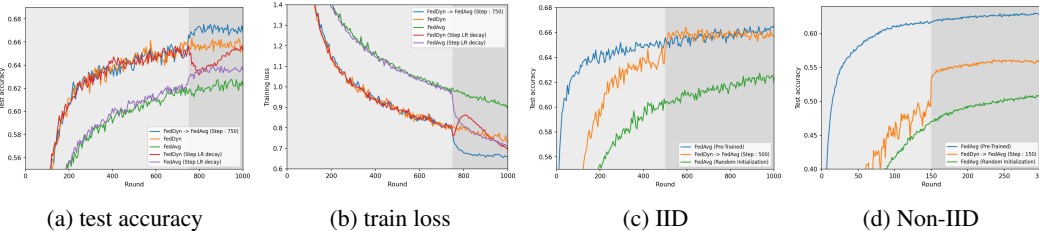

| (a) test accuracy | (b) train loss | (c) IID | (d) Non-IID |

Figure 7: The differences between learning rate decay and regularization-based initialization in terms of (a) test accuracy (b) train loss. The experimental results comparing pre-training using a public dataset and pre-training using regularization in (c) IID and (d) Non-IID settings.

initialization and final-training methods are possible. In this work, however, we restrict our analysis to FedDyn, Scaffold, FedNTD, MOON and the baseline FedAvg, leaving exploration of broader combinations to future work.

## 5 FURTHER ANALYSIS

### 5.1 COMPARISON WITH LEARNING RATE DECAY

In Figure 4, the training dynamics of regularization-based initialization resemble those of step-decay learning rate schedules. This raises the question of whether careful learning rate decay could replicate the benefits of pre-training with regularization. To examine this, we compared our default schedule (decaying the learning rate by a fixed ratio each round) with a variant that additionally decays the learning rate by one tenth at the pre-training step in Algorithm 1. As shown in Figures 7a and 7b, the two approaches produce markedly different dynamics. Thus, simple learning rate adjustment cannot substitute for initialization through regularization.

### 5.2 COMPARISON TO PRE-TRAINING ON PUBLIC DATASET

Prior work on pre-training in federated learning (Chen et al., 2023; Nguyen et al., 2023) demonstrates that data heterogeneity can be alleviated by initializing with a model pre-trained on a large external dataset. While it is not directly comparable, since our approach relies solely on client data whereas theirs leverages a public dataset, we conducted experiments to gauge the upper bound of our method. Using CIFAR-100, we compared FedDyn-based pre-training with models pre-trained for five epochs on a downsampled ImageNet-1k dataset. Results are shown in Figure 7. In the IID case, despite faster convergence with ImageNet pre-training, the final test accuracy is similar to that achieved with FedDyn, highlighting the strong initialization effect of regularization. In contrast, under non-IID distributions, the performance gap becomes substantial, indicating that initialization plays an even more critical role when data is heterogeneous.

## 6 CONCLUSION AND FUTURE DIRECTION

While many methods have been proposed to mitigate data heterogeneity in federated learning, it remains unclear what distinct characteristics these methods truly exhibit. Prior work has largely focused on theoretical convergence rates, whereas this study examined algorithms from the perspective of clients' pseudo-gradients and their impact on the features learned by the model. Our analysis shows that regularization is not universally effective; instead, it is more beneficial as a pre-training strategy for FedAvg than as a standalone approach. Because data heterogeneity is closely tied to initialization, we argue that FL algorithms should be analyzed more carefully from the perspectives of initialization and fine-tuning. We acknowledge that our work does not settle the broader question of where different regularization methods ultimately converge or which convergence point is optimal. Rather, our aim was to challenge the assumption that regularization is always beneficial throughout the entire training process for addressing heterogeneity.

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

## A APPENDIX

### A.1 DIFFERENCE BETWEEN SERVER CONTROL VARIATES AND GLOBAL GRADIENTS

As discussed in Section 3.4, the server control variate $h^t$ does not accurately approximate the global gradient. To further investigate its effect, we analyze in Figure 8 the impact of applying a single step update using $h^t$. As shown in Figure 8, the global model $\theta^t$ obtained by taking a single step from $\gamma^t$ with $h^t$ performs worse than the aggregation result $\gamma^t$ itself. This tendency is more pronounced in cross-silo settings than in cross-device settings, which contributes to FedDyn performing worse than FedAvg in the cross-silo scenario. Thus, in FedDyn, $h^t$ cannot be regarded as a reliable approximation of the global gradient.

### A.2 MORE ABLATION STUDY

**Switch point.** We present the results of an ablation study on the switch point. In the Table 2, we report three metrics: (1) the final accuracy of the global model, (2) the number of rounds required to reach the target accuracy - 67%, and (3) the total computational cost, all evaluated with respect to different switch points. We conducted federated learning for a total of 1000 rounds. When the

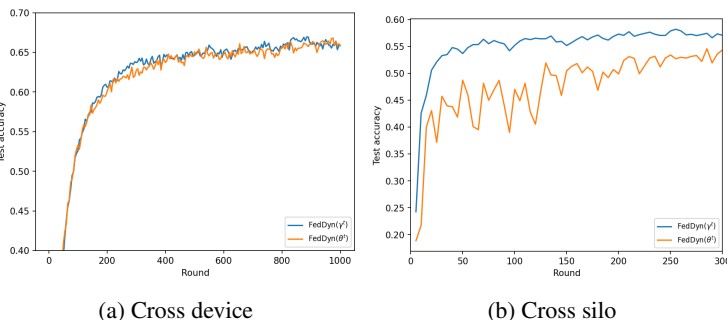

(a) Cross device          (b) Cross silo

Figure 8: Comparison of test accuracy: $\gamma^t$ (aggregation) vs. $\theta^t = \gamma^t - \frac{1}{\alpha} h^t$ (global model).

switch round is set to 0, the method corresponds to FedAvg, whereas setting the switch round to 1000 results in FedDyn. As the switch round increases incrementally by 150 rounds, we observe a corresponding increase in the final accuracy. If the target accuracy for this federated learning setup is 67%, regularization should be maintained for at least 600 rounds. According to the benchmark results reported in Baumgart et al. (2024), the computational cost of FedDyn compared to FedAvg is at least 30% higher and can reach up to 300%, depending on the hardware type and model architecture. For simplicity, we assume that the computational cost of FedDyn is 50% higher than that of FedAvg, and we estimate the expected computational overhead of the algorithm based on the switch point. As shown in the table, delaying the switch to FedAvg results in higher final accuracy. However, this comes at the cost of increased computational overhead and a greater number of rounds required to reach the target accuracy. This highlights a trade-off between final accuracy and efficiency (in terms of both computation cost and convergence speed). Therefore, the server should choose the switch round based on specific requirements or constraints.

Table 2: Ablation study on the switch point: comparison of final accuracy, rounds to reach target accuracy (67%), and relative computation cost.

| Switch round | 0 (FedAvg) | 1000 (FedDyn) | 150 | 300 | 450 | 600 | 750 | 900 |
|---|---|---|---|---|---|---|---|---|
| Final accuracy | 62.58 | 65.71 | 63.87 | 64.98 | 65.94 | 66.99 | 67.13 | 67.71 |
| Rounds to 67% | x | x | x | x | x | 660 | 790 | 910 |
| Computation cost | 1 | 1.5 | 1.075 | 1.15 | 1.225 | 1.3 | 1.375 | 1.45 |

**Effect of $\gamma_{\text{corr}}$ and $\gamma_{\text{data}}$.** We set $\gamma_{\text{corr}} = 0.5$ and define $\gamma_{\text{data}}$ as the 90th percentile of all entries in normalized $v_{(m,k)}$. Here, $\gamma_{\text{corr}}$ defines the threshold for determining whether two features should be considered the same, while $\gamma_{\text{data}}$ controls how many data samples contain specific features. Since our analysis does not focus on the frequency of feature occurrence across data samples, we conducted an ablation study only on $\gamma_{\text{corr}}$, examining its effect on the total number of feature clusters. The results in Table 3 indicate that as $\gamma_{\text{corr}}$ increases, the total number of feature clusters also increases. This is because a larger $\gamma_{\text{corr}}$ imposes a stricter criterion, making it harder for two features to be regarded as the same.

Table 3: Effect of varying $\gamma_{\text{corr}}$ on the number of feature clusters.

| | $\gamma_{\text{corr}} = 0.3$ | $\gamma_{\text{corr}} = 0.5$ | $\gamma_{\text{corr}} = 0.7$ |
|---|---|---|---|
| FedAvg | 47 | 142 | 206 |
| FedDyn | 27 | 105 | 198 |

A.3 OTHER METHODS FOR FINE-TUNING

We evaluated other regularization-based methods such as FedNTD and Scaffold to assess their suitability for the second stage of training under a cross-device non-IID distribution using the CIFAR-100 dataset. We excluded MOON from our comparison because it requires modifications to the model architecture. As shown in Table 4, FedNTD is not effective during the second stage of training, whereas Scaffold performs comparably to FedAvg. Previous studies have demonstrated that the performance of federated learning algorithms does not differ significantly when models are properly initialized. Therefore, we argue that it is sufficient to use FedAvg in the second stage, as it introduces no additional computational or communication overhead.

Table 4: Performance of different methods in the second stage of training under cross-device non-IID CIFAR-100.

| Method | FedDyn (continuous) | FedAvg | FedNTD | Scaffold |
|---|---|---|---|---|
| Accuracy | 54.25 | 56.48 | 54.06 | 56.31 |

A.4 PROOF OF SWITCHING POINT CONDITION

In this appendix, we provide the detailed derivation leading to Proposition 1. Our goal is to analyze the effect of the server update vector $h^t$ in FedDyn and establish conditions under which continued training with FedDyn ceases to be beneficial, thereby motivating a switch to FedAvg.

**Preliminaries.** We build upon Lemma 1 of (Acar et al., 2021), which states:

$$\kappa_0 E\big[\ell(\gamma^{t-1}) - \ell(\theta_*)\big] \leq \Phi_{t-1} - \Phi_t, \tag{6}$$

where

$$\gamma^t = \frac{1}{P} \sum_{k \in P_t} \theta_k^t, \qquad \Phi_t = E\|\gamma^t - \theta_*\|^2 + \kappa C_t.$$

**Bounding the Effect of $h^t$.** To analyze the effect of the server update $h^{t-1}$, we expand:

$$\ell(\theta^{t-1}) = \ell\big(\gamma^{t-1} - \tfrac{1}{\alpha} h^{t-1}\big) \tag{7}$$

$$\leq \ell(\gamma^{t-1}) - \tfrac{1}{\alpha}\langle \nabla\ell(\gamma^{t-1}), h^{t-1}\rangle + \tfrac{L}{2\alpha^2}\|h^{t-1}\|^2 \tag{8}$$

$$\leq \ell(\gamma^{t-1}) + \tfrac{1}{2\alpha}\|\nabla\ell(\gamma^{t-1})\|^2 + \tfrac{1}{2\alpha}\|h^{t-1}\|^2 + \tfrac{L}{2\alpha^2}\|h^{t-1}\|^2 \tag{9}$$

$$= \ell(\gamma^{t-1}) + \tfrac{1}{2\alpha}\|\nabla\ell(\gamma^{t-1}) - \nabla\ell(\theta_*)\|^2 + \big(\tfrac{1}{2\alpha} + \tfrac{L}{2\alpha^2}\big)\|h^{t-1}\|^2 \tag{10}$$

$$\leq \ell(\gamma^{t-1}) + \tfrac{L}{\alpha}\big(\ell(\gamma^{t-1}) - \ell(\theta_*)\big) + \big(\tfrac{1}{2\alpha} + \tfrac{L}{2\alpha^2}\big)\|h^{t-1}\|^2. \tag{11}$$

The first inequality follows from the descent lemma (Eq. (4) in (Acar et al., 2021)). The second inequality follows from $\|\nabla\ell(\gamma^{t-1}) + h^{t-1}\|^2 \geq 0$. The third uses $\nabla\ell(\theta_*) = 0$, and the last inequality uses $\tfrac{1}{2L}\|\nabla\ell(\gamma^{t-1})\|^2 \leq \ell(\gamma^{t-1}) - \ell(\theta_*)$.

**Upper Bound on the Expected Loss.** Taking expectation and applying Lemma 1 yields:

$$E[\ell(\theta^{t-1}) - \ell(\theta_*)] \leq \tfrac{(1+\frac{L}{\alpha})}{\kappa_0}(\Phi_{t-1} - \Phi_t) + \big(\tfrac{1}{2\alpha} + \tfrac{L}{2\alpha^2}\big) C_{t-1}. \tag{12}$$

Summing over $T$ rounds, we obtain:

$$\sum_{t=1}^{T} E[\ell(\theta^{t-1}) - \ell(\theta_*)] \leq \tfrac{(1+\frac{L}{\alpha})(\Phi_0 - \Phi_T)}{\kappa_0} + \big(\tfrac{1}{2\alpha} + \tfrac{L}{2\alpha^2}\big)\sum_{t=1}^{T} C_{t-1}. \tag{13}$$

Dividing by $T$, we define the cumulative upper bound:

$$D_T = \tfrac{1}{T}\left(\zeta + \eta\sum_{t=1}^{T} C_{t-1}\right), \tag{14}$$

where

$$\zeta = \big(1 + \tfrac{L}{\alpha}\big)\tfrac{\Phi_0}{\kappa_0}, \qquad \eta = \tfrac{1}{2\alpha} + \tfrac{L}{2\alpha^2}.$$

**Switching Condition.** We identify the switching point $T$ as the smallest index such that $D_T \leq D_{T+1}$, i.e.,

$$T\eta\Big(C_T - \tfrac{1}{T}\sum_{t=1}^{T} C_{t-1}\Big) \geq \zeta. \tag{15}$$

This condition formalizes the tradeoff between the benefit of regularization and the cost induced by server-side compensation. When $C_t$ is small, FedDyn remains beneficial; when $C_t$ becomes large, switching to FedAvg is preferable.

**Discussion.** Two remarks are in order: 1. The derivation uses $\|\nabla\ell(\gamma^{t-1}) + h^{t-1}\|^2 \geq 0$. Since $h^{t-1}$ partially approximates the global gradient, this may appear loose. Empirically, however, $\gamma^t$ achieves higher accuracy than $\theta^t$ in most cases, suggesting that $h^t$ indeed fails to capture the global gradient, making the bound meaningful. 2. The inequality requires $C_T - \frac{1}{T}\sum_{t=1}^{T} C_{t-1} \geq 0$. Because $h^t$ does not approximate the global gradient uniformly across rounds, $C_t$ need not decrease monotonically. Moreover, FedDyn enforces updates orthogonal to past gradients, making it plausible for $C_T$ to exceed its historical average.

## A.5 ALGORITHM

---

**Algorithm 1** FedDyn to FedAvg

---

Initialization : $\theta^0, h^0 = \mathbf{0}, \nabla L_k(\theta_k^0) = 0$

**Server executes:**

**for** round $t = 0, 1, \ldots T-1$ **do**

  $P_t \leftarrow$ Random Clients

  **for** each client $k \in P_t$, and in parallel **do**

    $\theta_k^{t+1} \leftarrow$ **Client_Update**$(k, \theta^t, t)$

  **end for**

  $\theta^{t+1} = \frac{1}{|P_t|}\sum_{k \in P_t} \theta_k^{t+1}$

  **if** t < Pre-training Step **then**

    $h^{t+1} = h^t - \alpha\frac{1}{m}\sum_{k \in P_t} (\theta_k^{t+1} - \theta^t)$

    $\theta^{t+1} = \theta^{t+1} - \frac{1}{\alpha} h^{t+1}$

  **end if**

**end for**

**Client_Update**$(k, \theta^t, t)$**:**

$\theta_k^{t+1} \leftarrow \theta^t \quad \nabla L_k(\theta_k^t) \leftarrow \nabla L_k(\theta_k^{latest\_updated})$

**for** local epoch $e = 1, 2, \ldots E$ **do**

  **for** each mini batch **b** **do**

    **if** t < Pre-training Step **then**

      $L(\theta) = L_k(\theta) - \langle\nabla L_k(\theta_k^t), \theta\rangle + \frac{\alpha}{2}||\theta - \theta^t||^2$

    **end if**

    $\theta_k^{t+1} \leftarrow \theta_k^{t+1} - \eta\nabla L(\theta_k^{t+1}; \mathbf{b})$

  **end for**

**end for**

$\nabla L_k(\theta_k^{t+1}) \leftarrow \nabla L_k(\theta_k^t) - \alpha(\theta_k^{t+1} - \theta^t)$

return $\theta_k^{t+1}$

---

## A.6 ALPHA SENSITIVITY

FedDyn utilizes client drift $\nabla L_k(\theta_k^t)$ to regularize each client's local update, and the server updates the global model using $h$, which is the average of each client's local drift. FedDyn's results are sensitive to $\alpha$, and hyperparameter search is necessary to find an appropriate $\alpha$ value. The $\alpha$ affects the scale of local drift $\nabla L_k(\theta_k^t)$, but when updating using $h$ on the server, it is not affected by the $\alpha$ because it is rescaled by $\frac{1}{\alpha}$. In other words, the $\alpha$ value only affects the scale of local drift, and a large $\alpha$ means stronger regularization. We experimented with how the results of FedDyn → FedAvg change as the alpha value changes. We experimented with the results of FedDyn → FedAvg with varying the $\alpha$ value.

As shown in Figure 9, the results of FedDyn→ FedAvg vary depending on the $\alpha$ value. If $\alpha$ is too large, such as 1.0, it is worse than FedAvg, and when $\alpha$ is 0.1 or 0.01, the performance of FedDyn is similar regardless of the $\alpha$ value, but the performance after fine-tuning with FedAvg is different depending on the $\alpha$ value. We also experimented with how pseudo gradient diversity and local drift diversity change depending on the $\alpha$ value.

As shown in Figure 10, the smaller $\alpha$, the higher the pseudo gradient diversity and the greater the local drift diversity. Conversely, as the $\alpha$ increases, gradient diversity decreases, but the converge speed is very slow due to excessive regularization. When the $\alpha$ value was 1.0, FedDyn→FedAvg had no performance gain unlike other $\alpha$ values, which is presumed to be due to low local drift diversity. However, it seems very difficult to find the perfect $\alpha$ value that achieves fast converge while reducing local drift diversity, and it is unknown whether it exists. Instead, we propose a two-stage learning

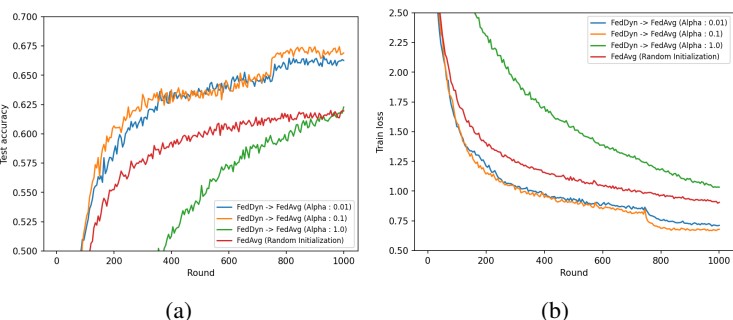

(a)                                            (b)

Figure 9: Test accuracy (a) and train loss (b) according to $\alpha$ value (CIFAR100: IID, 10% participation)

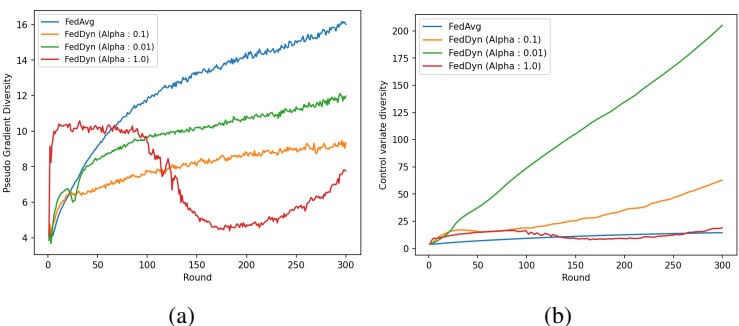

(a)                                            (b)

Figure 10: Pseudo gradient diversity (a) and local drift diversity (b) according to $\alpha$ value (CIFAR100: nonIID, Full participation)

method that first converges to a global stationary point where the summation of local drift becomes 0 through FedDyn and then fine-tunes with FedAvg.

### A.7 ANALYSIS OF DYNAMIC REGULARIZATION

So, why is dynamic regularization, which reduces client drift by inducing updates in a direction orthogonal to the direction in which the client has updated so far, not always effective, and why is it converging to a good initialization point for FedAvg? To answer this question, we analyzed FedDyn's training dynamics in more detail. First, FedDyn defines local drift $\nabla L_k(\theta_k^{t+1})$ as the accumulation of pseudo gradient $\theta_k^{t+1} - \theta^t$ and uses the local drift to regularize local updates. In the FedDyn paper, this definition of local drift is defined through the first order condition for local optima, but there is no guarantee that the first order condition for local optima will be satisfied during actual local update. What can be inferred from the subsequent experimental result is that local drift, defined as the accumulation of pseudo gradient, is meaningful in itself even if it is not justified as a first order condition for local optimality. Second, the server defines $h$ state as the average of the local drift $\nabla L_k(\theta_k^{t+1})$ of clients, and then additionally updates the global model using the $h$ after aggregation of local models. Regularization using $\nabla L_k(\theta_k^{t+1})$ and global model update using $h$ are complementary to each other. FedDyn converges to the stationary point of global risk because the converge of $\theta^t$ implies $h = \sum_k \nabla L_k(\theta_k^{t+1}) \to 0$.

However, we found that FedDyn achieves $\sum_k \nabla L_k(\theta_k^{t+1}) \to 0$ by merely increasing the diversity of local drift $\nabla L_k(\theta_k^{t+1})$. We measured the diversity of the local drift $\nabla L_k(\theta_k^{t+1})$ of each client along Equation (1) as well as the pseudo gradient. In Figures 11a and 11b, the diversity of local drift is much larger than the diversity of pseudo gradient. In particular, when the number of clients is 10, the diversity of local drift is extremely high compared to the diversity of pseudo gradient, and this is assumed to be the reason for the poor performance of FedDyn when the number of clients is small. However, FedDyn makes the global model quickly converge to the stationary of global risk, and this stationary point seems to be a good initialization point for FedAvg. We conducted an ablation study

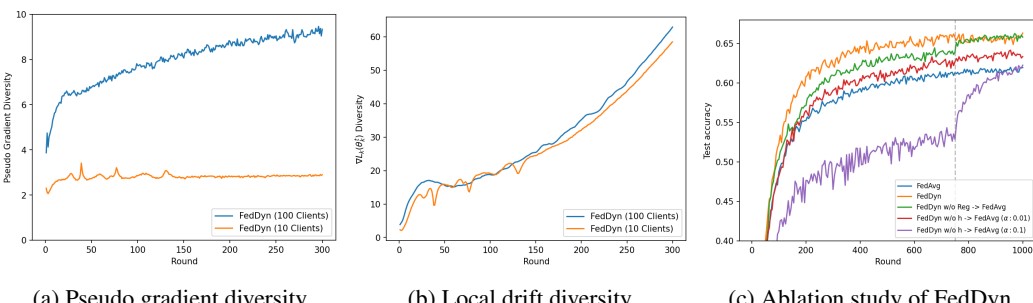

(a) Pseudo gradient diversity     (b) Local drift diversity     (c) Ablation study of FedDyn

Figure 11: Training dynamics of FedDyn.

to analyze which of regularization using local drift on the client side and update using $h$ on the server side contributes to increasing the diversity of local drift. As shown in the Section A.8, update on the server side using $h$ contributes to increasing the diversity of local drift, and in Figure 11c, even without regularization, update on the server side using $h$ helps fast converge of the global model. Note that in the absence of regularization, local drift $\nabla L_k(\theta_k^{t+1})$ is not derived from the first order condition of local optimality, but is simply the accumulation of the pseudo gradient. Considering that the local drift of each client is the accumulation of the pseudo gradient in every round, and that $h$ on the server is the summation of local drift, an update using $h$ on the server is a type of server momentum (Hsu et al., 2019). Conversely, when regularizing without updating using $h$ on the server side, it is effective only when the $\alpha$ is small, that is, when regularization is applied weakly. Note that FedDyn w/o Reg $\rightarrow$ FedAvg is similar to the performance of FedDyn without any additional computational overhead unlike FedDyn.

## A.8 ABLATION STUDY OF FEDDYN

Let's call FedDyn w/o Reg an algorithm that does not apply regularization in FedDyn, but calculates local drift $\nabla L_k(\theta_k^t)$ and $h$ in the same way and updates the global model using $h$ on the server. Fed-Dyn w/o Reg is identical to FedDyn with $\alpha \rightarrow 0$. Conversely, let's call FedDyn w/o h an algorithm that only applies dynamic regularization in FedDyn and does not update using h. While FedDyn w/o Reg does the same local update as FedAvg, FedDyn w/o h has the same level of computational overhead as FedDyn. As shown in Figure 11c, FedDyn w/o Reg $\rightarrow$ FedAvg is effective enough to provide similar performance to FedDyn without any overhead. We analyzed whether the reason FedDyn$\rightarrow$FedAvg is effective is because regularization has ended or because there are no longer updates using $h$.

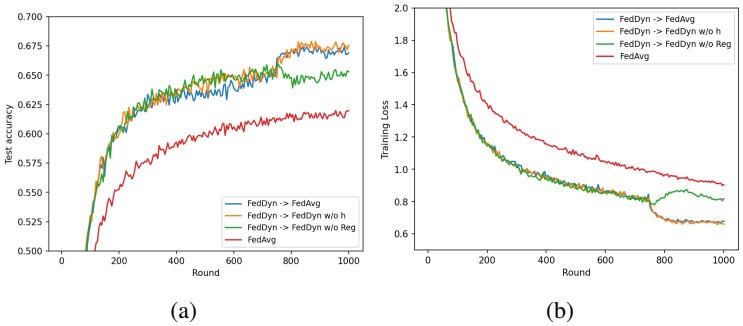

(a)          (b)

Figure 12: Test accuracy (a) and train loss (b) with fine-tuning with Modified FedDyn (CIFAR100: IID, 10% participation)

As shown in Figure 12, FedDyn$\rightarrow$Fed w/o Reg is not much different from continuing to use FedDyn. In contrast, FedDyn$\rightarrow$FedDyn w/o h shows almost similar performance to FedDyn$\rightarrow$FedAvg. We tried to analyze the reason through pseudo gradient diversity and local drift diversity.

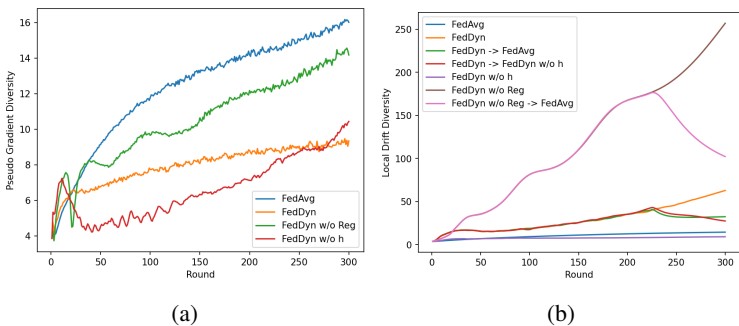

(a)  (b)

Figure 13: Pseudo gradient diversity (a) and local drift diversity (b) of Modified FedDyn (CIFAR100: nonIID, Full participation)

As shown in Figure 13, FedDyn w/o h shows lower pseudo gradient diversity and also lower local drift diversity compared to FedDyn. However, FedDyn w/o h has very slow convergence because there is no compensation for regularization. In contrast, FedDyn w/o Reg has higher pseudo gradient diversity than FedDyn and also has higher local drift diversity. The high local drift diversity of FedDyn w/o Reg explains the performance gain of FedDyn w/o Reg→FedAvg. In summary, regularization lowers pseudo gradient diversity, but requires compensation using $h$ on the server, which in turn increases local drift diversity, falling into the stationary point of the global objective. And this stationary point is a good initialization point for FedAvg.

## A.9 HYPERPARAMETER SETTING

Table 5: Hyperparameters and model architecture used in Cross-device experiments

| Data | CIFAR10 | CIFAR100 | Tiny-ImageNet |
|---|---|---|---|
| Model | VGG-11 | Resnet-20 | Resnet-34 |
| Hidden size | [64, 128, 256, 512] | [16, 32, 64] | [64, 128, 256, 512] |
| Client $N$ | | 100 | |
| Participation ratio $C$ | | 10% | |
| Local Epoch $E$ | | 5 | |
| Local Batch Size $B$ | | 50 | |
| Optimizer | | SGD | |
| Momentum | | 0. | |
| Weight decay | | 1e-3 | |
| $\alpha$ (FedDyn) | | 0.1 | |
| Communication rounds | | 1000 | |
| Learning rate $\eta$ | | 0.1 | |
| Learning rate decay | | 0.999 | |

Table 6: Hyperparameters and model architecture used in Cross-silo experiments

| Data | CIFAR10 | CIFAR100 | Tiny-ImageNet |
|---|---|---|---|
| Model | VGG-11 | Resnet-20 | Resnet-34 |
| Hidden size | [64, 128, 256, 512] | [16, 32, 64] | [64, 128, 256, 512] |
| Client $N$ | | 10 | |
| Participation ratio $C$ | | 100% | |
| Local Epoch $E$ | 1 | 5 | 1 |
| Local Batch Size $B$ | | 50 | |
| Optimizer | | SGD | |
| Momentum | | 0. | |
| Weight decay | | 1e-3 | |
| $\alpha$ (FedDyn) | | 0.01 | |
| Communication rounds | | 300 | |
| Learning rate $\eta$ | | 0.1 | |
| Learning rate decay | | 0.99 | |

