# OpenReview forum: "Rethinking Regularization in Federated Learning: An Initialization Perspective"
_ICLR.cc/2026/Conference — Submitted to ICLR 2026_

### Official Review · Reviewer_NfTo · 2025-10-25

**Soundness:** 2
**Presentation:** 2
**Contribution:** 3
**Rating:** 4
**Confidence:** 3

**Summary:**

This paper points out that existing SoTA methods, such as FedDyn, may impose too strong the regularization that impedes from reaching better global minima. The authors propose an intriguing trick: switch back to vanilla FedAvg. With analysis from gradient diversity, cosine similarity, and feature cluster analysis, this work shows improvement over constrained setup (only Table 1 and Figure 1 seems to be serious quantitative results to me). I encourage the authors to provide additional experiments to better understand this algorithm-switch phenomenon to better understand and strengthen their observations.

**Strengths:**

1. The algorithm-switch phenomenon is interesting and Figure 1 is specifically convincing to me.
2. The perspective and comparison with learning decay is important to ensure the performance gain is not similar to learning rate scheduler.

**Weaknesses:**

1. Limited algorithm choice

All the main quantitative analysis focuses on FedDyn, and there are no metrics regarding FedNTD or Scaffold. The authors also should consider more recent algorithms such as FedAlign, FedExp, FedProto, FedMD, or others. It is hard to demonstrate how robust, consistent, and reproducible this method is empirically without a thorough comparison. I strongly encourage the authors to include an additional quantitative table comparing at least three additional federated learning algorithms using the same setup as in Table 1.

2. What is the main contribution of algorithm-switching?

I am not fully convinced that "regularization impedes global feature learning" is the best explanation for the working mechanism of the proposed method. My intuition is that changing (or switching) the minimization goal prevents the model from getting stuck at a local minimum, because that is effectively what the proposed method is doing. I would appreciate it if the authors could conduct additional experiments to prove this hypothesis right or wrong. For example, instead of switching from FedDyn to FedAvg, can the authors try switching from FedAvg to FedDyn? Another interesting setup would be to switch between FedDyn and FedAvg every 20 rounds and evaluate the performance.

**Questions:**

1. Can the authors explain why in Figure 1b there are some abrupt drops of diversity but in Figure 1c the cosine similarities increase monotonically?

---

> ### Author Response · Authors · 2025-11-21
> **Response to reviewer NfTo (1/2)**
>
> We thank the reviewer for the constructive and insightful comments. We address each concern in detail below.
>
> **W1. Limited algorithm choice. The authors also should consider more recent algorithms such as FedAlign, FedExp, FedProto, FedMD, or others.**
>
> In response to the reviewer’s feedback, we evaluated the proposed FedDyn${}\rightarrow{}$FedAvg strategy against additional FL algorithms, including FedAlign[1], FedDecorr[2], and FedFA[3].
> (Other methods suggested by the reviewer, such as FedExp, FedProto, and FedMD, are not client regularization–based aggregation methods, but instead target server step-size adaptation or heterogeneous model architectures; thus they were excluded.)
>
> For a fair comparison, we reproduced the FedAvg performance reported in each paper and used these reproduced results to reconstruct their experimental settings.
> We then used the baseline results from the original tables and evaluated FedDyn and our two-stage FedDyn${}\rightarrow{}$FedAvg method under the same settings.
>
> Across these benchmarks, FedDyn consistently outperforms the other algorithms, and the two-stage strategy provides further improvements.
> We report the FedAvg results from the FedAlign paper (with our reproduced numbers in parentheses), while the remaining results for FedAlign, FedProx, and MOON are taken directly from the FedDecorr paper.
>
> | FedAvg¹ | FedProx¹ | MOON¹ | FedAlign¹ | FedDyn | FedDyn → FedAvg |
> |---------|----------|--------|------------|---------|------------------|
> | 52.9 (53.0) | 53.1 | 55.0 | 56.9 | (57.7) | (59.2) |
>
> We compare FedDyn and our proposed two-stage training strategy with FedDecorr.
> As shown in the table, FedDyn consistently outperforms other regularization-based methods, including FedDecorr, and its advantage becomes even more pronounced as the non-IID level increases.
> Moreover, our two-stage training strategy further improves performance and consistently surpasses applying regularization throughout the entire training process.
>
> | Method                   | α=0.05         | 0.1            | 0.5            | ∞              |
> |--------------------------|----------------|----------------|----------------|----------------|
> | FedAvg²                  | 59.87 (60.39)  | 66.46 (66.32)  | 71.69 (71.94)  | 74.54 (74.26)  |
> | +FedDecorr²              | 61.53          | 67.12          | 71.91          | 73.87          |
> | Scaffold²                | 54.51          | 61.42          | 68.37          | 70.97          |
> | MOON²                    | 56.97          | 65.48          | 71.81          | 74.30          |
> | FedDyn                   | (64.40)        | (68.06)        | (72.31)        | (74.74)        |
> | FedDyn → FedAvg          | (65.38)        | (69.55)        | (74.27)        | (76.29)        |
>
>
> We also compared our approach with FedFA.
> However, reproducing the results reported in Table 1 of the FedFA paper was considerably more challenging.
> The FedFA setup disables data augmentation and restricts training to only 200 communication rounds, leaving insufficient time for models to converge.
> Under such conditions, the benefit of our two-stage strategy (FedDyn${}\rightarrow{}$FedAvg) is limited because FedDyn cannot reach its global stationary point before the switch.
> Additionally, several hyperparameters in the FedFA paper are not clearly specified, making exact reproduction of their FedAvg results difficult.
> Nevertheless, we approximated the FedAvg baseline and trained FedDyn under identical conditions.
> In this setting, the FedDyn$\rightarrow$FedAvg switch is less effective because FedDyn does not have enough rounds to fully reach its stationary behavior before transitioning.
>
> | Method               | α=0.1         | 0.5            | ∞              |
> |----------------------|----------------|----------------|----------------|
> | FedAvg³              | 21.79 (21.25)  | 26.52 (25.04)  | 25.37 (26.81)  |
> | FedFA³               | 24.05          | 29.16          | 33.94          |
> | MOON³                | 22.04          | 26.69          | X              |
> | FedProx³             | 22.30          | 26.03          | X              |
> | FedDyn               | (34.36)        | (35.69)        | (33.89)        |
> | FedDyn → FedAvg      | (33.10)        | (35.70)        | (34.28)        |
>
>
> Through comparisons with other federated learning algorithms, we observe that FedDyn achieves superior performance even under the experimental settings optimized for each respective method.
> Moreover, in most cases, the proposed two-stage strategy (FedDyn${}\rightarrow{}$FedAvg) also demonstrates strong and consistent effectiveness.
>
> [1] Local Learning Matters: Rethinking Data Heterogeneity in Federated Learning. CVPR 22
> [2] Towards Understanding and Mitigating Dimensional Collapse in Heterogeneous Federated Learning. ICLR 23
> [3] FedFA: Federated Learning With Feature Anchors to Align Features and Classifiers for Heterogeneous Data. TMC 24

---

> > ### Comment · Reviewer_NfTo · 2025-11-21
> >
> > I thank the reviewer for the effort in conducting additional experiments.
> >
> > I think I did not make it clear about [W1]. What I am interested in is not including "baselines" but including baseline experiments such as "MOON → FedAvg" or "FedProx → FedAvg" or even "FedDyn → MOON" (or other combinations). It would be a strong rebuttal if the authors could provide them.
> >
> > The reason I mention or demand extra experiments relates to how I think about the contribution of this paper. I read your rebuttal, as in Response to reviewer NfTo (2/2), and appreciate the explanation. However, I am not interested in whether "FedDyn → FedAvg" can beat "FedAvg  →FedDyn". I am curious about whether the method, i.e., switching algorithms, achieves its magic by changing the loss landscape for the algorithm and thus enabling the avoidance of saddle points or local minima, resulting in better results.
> >
> > I think this is a very strong contribution compared to its current form. Currently, your explanation of mechanisms makes sense, but it narrows the contribution and makes it a niche, cause it only works on FedDyn. And I am not sure if it is worth pursuing for acceptance, as per the reviewer PREy concerns.

---

> ### Author Response · Authors · 2025-11-21
> **Response to reviewer NfTo (2/2)**
>
> **W2. What is the main contribution of algorithm-switching? For example, instead of switching from FedDyn to FedAvg, can the authors try switching from FedAvg to FedDyn? Another interesting setup would be to switch between FedDyn and FedAvg every 20 rounds and evaluate the performance.**
>
> The experimental results for the reverse direction, FedAvg$\rightarrow$FedDyn, are presented below.
>
> | Metric        | FedAvg | FedDyn | FedDyn → FedAvg | FedAvg → FedDyn |
> |---------------|--------|--------|------------------|------------------|
> | Accuracy (%)  | 62.58  | 65.71  | 67.18           | 64.47           |
>
>
> The experimental results indicate that FedAvg does not converge to a good initialization point for FedDyn.
> Practically, regardless of the final accuracy, the fast-convergence properties of FedDyn cannot be exploited when training begins with FedAvg. **The key factor behind the effectiveness of the FedDyn$\rightarrow$FedAvg strategy is not the act of switching algorithms itself, but rather the fact that FedDyn converges to a point near the global stationary point.**
>
> Among the methods we tested, FedDyn was also the only approach that demonstrated clear potential as an initialization mechanism for federated learning. We do not claim that all existing regularization-based methods converge to good initialization points in federated learning.
> However, **if an ideal form of regularization were able to perfectly align the local objectives and guide the optimization trajectory toward the true global stationary point, analogous to pre-training on high-quality public data, this would naturally yield a strong initialization.**
> From this viewpoint, FedDyn is unique in that it asymptotically aligns the local and global objectives, and our experiments confirm this through consistent patterns observed in gradient diversity, feature learning behavior, and other analytical metrics.
>
> **Q1. Can the authors explain why in Figure 1b there are some abrupt drops of diversity but in Figure 1c the cosine similarities increase monotonically?**
>
> In the case of Figure 1(c), we measure the similarity of pseudo gradients within the same client.
> Because the method switches to FedAvg, which does not apply any regularization, the pseudo gradients naturally become more similar to the final round's pseudo gradient, explaining the overall increase in similarity.
> In contrast, Figure 1(b) measures the diversity of pseudo gradients across different clients.
> The sudden change in the objective function at the switching point appears to induce a temporary fluctuation in this diversity.

---

> ### Author Response · Authors · 2025-11-27
>
> We have clearly understood the reviewer's concerns.
> The key question is whether switching algorithms (not only from FedDyn → FedAvg but also other combinations) can reshape the loss landscape so that the model escapes local minima and improves performance.
> Based on the additional experiments below, our response is summarized as follows.
>
> As noted by the reviewer, switching algorithms indeed changes the loss landscape, which substantially affects training, as shown by the FedAvg → FedDyn example in the previous rebuttal.
> However, such changes do not always guarantee better performance.
> **This motivation arises from prior works [1,2], which show that data heterogeneity in federated learning is strongly linked to initialization.**
> When training starts from well-initialized weights, heterogeneity has less impact, and performance differences between algorithms become marginal.
>
> Our core idea follows this insight:
> **if a regularization method can mitigate heterogeneity and converge to a global stationary point, its convergence point should correspond to the “pre-trained initialization” described in [1,2].**
> In such cases, switching to any other algorithm should yield consistent results, regardless of the specific choice of algorithm.
> We therefore explore not only the FedDyn → FedAvg case but also various algorithm combinations.
>
> We tested other regularization-based methods (FedNTD, Scaffold) for the second training stage under a cross-device non-IID CIFAR100 setup.
> MOON was excluded as it requires model modifications.
> As shown below, the choice of algorithm in the second stage has little impact on performance, supporting the use of FedAvg for its simplicity and zero additional overhead.
>
> | Method  | FedAvg | FedDyn | FedDyn → FedAvg | FedDyn → FedNTD | FedDyn → Scaffold |
> | :-------: | :------: | :------: | :---------------: | :---------------: | :-----------------: |
> | Acc (%) | 53.51  | 54.25  | 56.48       | 56.24           | 56.31             |
>
>
> We also examined initialization effects in other algorithms.
> Scaffold, with similar motivation, was tested under a cross-device IID setting.
>
> | Method  | FedAvg | Scaffold | Scaffold → FedAvg | Scaffold → FedDyn |
> | :-------: | :------: | :--------: | :-----------------: | :-----------------: |
> | Acc (%) | 62.58  | 65.47    | 65.04             | 64.86             |
>
>
> These results suggest Scaffold does not yield an ideal initialization point, and the gain from late-stage regularization is limited, especially given its higher communication and computation costs.
> A similar trend holds for MOON and FedNTD.
>
> | Method  | FedAvg | MOON  | MOON → FedAvg | FedNTD | FedNTD → FedAvg |
> | :-------: | :------: | :-----: | :-------------: | :------: | :---------------: |
> | Acc (%) | 62.58  | 62.13 | 61.88         | 63.06  | 62.99           |
>
>
> In summary, our goal was to conduct a comprehensive comparative analysis of various regularization methods to better understand how regularization influences data heterogeneity in federated learning.
> First, beyond final accuracy, we examined other perspectives such as gradient diversity and feature learning, concluding that dynamic regularization provides the most ideal behavior.
> Second, from the viewpoint of initialization, we show that only dynamic regularization converges to a point that alleviates data heterogeneity.
> **While we understand the reviewer’s concern that the proposed FedDyn → FedAvg strategy may seem niche, we believe our contribution lies not merely in the method itself but in the in-depth analysis of how various regularization schemes influence data heterogeneity from multiple perspectives.**
>
> In the previous rebuttal (Response to reviewer NfTo (1/2)), we noted that many methods aiming to mitigate data heterogeneity (e.g., FedProx, MOON, Scaffold) often perform worse than FedAvg.
> Given the wide variety of FL settings (cross-silo, cross-device, client number, local epochs, etc.), these regularization methods do not behave consistently. **Rather than assuming that any form of regularization would somehow alleviate data heterogeneity, our goal was to critically examine whether these methods truly regularize the optimization process and guide the model toward a meaningful global stationary point.**
>
> [1] Where to begin? On the impact of pre-training and initialization in federated learning. ICLR 2023.
> [2] On the importance and applicability of pre-training for federated learning. ICLR 2023.

---

### Official Review · Reviewer_PREy · 2025-10-30

**Soundness:** 2
**Presentation:** 2
**Contribution:** 2
**Rating:** 2
**Confidence:** 3

**Summary:**

This paper reframes the use of regularization in federated learning, arguing that its primary benefit is in the early training phase. Through novel gradient and feature-level analyses, the authors show that while FedDyn effectively reduces client drift, its benefits diminish over time and it introduces side effects. They propose a two-stage strategy: use FedDyn for pre-training to find a strong initialization, then switch to standard FedAvg for fine-tuning. Experiments show this FedDyn -> FedAvg approach often achieves superior accuracy and faster convergence than using either method alone, while also reducing computational overhead.

**Strengths:**

1. This paper shifts the focus from finding the "best" single algorithm to understanding when an algorithm is most beneficial, introducing a valuable "initialization vs. fine-tuning" paradigm.
2. This paper provides deep, quantitative insights into how regularization methods work and where they fail by considering more metric like gradient diversity and feature interaction tensors.
3. The proposed two-stage strategy is simple, effective, and computationally efficient.

**Weaknesses:**

1. Lack of Theoretical Proof: The paper lacks a formal convergence proof for the proposed two-stage FedDyn -> FedAvg algorithm, relying instead on empirical results and a non-practical switching criterion.
2. Switching Point: The effectiveness of the method depends critically on the switching point, which is chosen heuristically in the experiments. The paper offers no practical guidance on how to determine this crucial hyperparameter.
3. Limited Generality: The conclusion that only FedDyn serves as a good initializer is not fully explained, limiting the generality of the "regularization for initialization" principle to other methods.
4. This paper reads more like an experimental report than an academic paper. Its contribution and innovation seem somewhat weak for a conference like ICLR.

**Questions:**

1. How can the optimal switching point be determined in a principled and adaptive way for new tasks?
2. What is the fundamental reason that FedDyn succeeds as an initializer while a similar method like SCAFFOLD fails?
3. Is the two-stage training principle generalizable to other combinations of FL algorithms beyond FedDyn -> FedAvg?

---

> ### Author Response · Authors · 2025-11-21
> **Response to reviewer PREy (1/2)**
>
> We thank the reviewer for the constructive and insightful comments. We address each concern in detail below.
>
> **W2-Q1. Switching Point: The effectiveness of the method depends critically on the switching point, which is chosen heuristically in the experiments. The paper offers no practical guidance on how to determine this crucial hyperparameter. How can the optimal switching point be determined in a principled and adaptive way for new tasks?**
>
> As the reviewer noted, we used a fixed switching point because it already provides strong performance, and thus we did not emphasize the link between the theoretical and practical criteria.
> We will include a concise discussion of this connection in the revised manuscript.
> While Appendix A.2 presents a sensitivity analysis of the switching point, we summarize the key findings here and introduce a practical guideline for selecting the switching point based on our theoretical analysis.
> Assuming that FedDyn incurs 50\% higher computational cost than FedAvg, the results show that delaying the switch improves final accuracy but increases computation and the number of rounds needed to reach the target accuracy.
>
> | Switch round     | 0 (FedAvg) | 1000 (FedDyn) | 150   | 300   | 450   | 600   | 750   | 900   |
> |------------------|------------|---------------|-------|-------|-------|-------|-------|-------|
> | Final accuracy   | 62.58      | 65.71         | 63.87 | 64.98 | 65.94 | 66.99 | 67.13 | 67.71 |
> | Rounds to 67%    | x          | x             | x     | x     | x     | 660   | 790   | 910   |
> | Computation cost | 1          | 1.5           | 1.075 | 1.15  | 1.225 | 1.3   | 1.375 | 1.45  |
>
> However, determining an appropriate pre-training duration in practice is not straightforward.
> Building on the theoretical switching point, we therefore propose a practical guideline for selecting the pre-train step.
> **Although there is an inherent trade-off between computation and performance, our aim is to provide a principled criterion that ensures sufficient performance while avoiding unnecessary regularization.**
>
> Proposition 1 states that the switching point should occur when $C_t$ exceeds its historical average.
> Since $C_t$ cannot be computed directly, we instead use a practical proxy: the server control variate $||h_t||$ should surpass its historical average.
> Because $||h_t||$ starts from zero and fluctuates during the early stage of training, we compute a slope over a 50-round window and identify the first interval in which the slope becomes positive; this interval is designated as the switching point.
> When the learning rate decays by a factor $r$, the magnitude of $h_t$ decreases, making this interval more difficult to detect.
> To compensate for this effect, we approximate the hypothetical trajectory under a constant learning rate.
> Consider the update rule $ h_{t+1} = h_t + n_t \delta,  n_t = n_0 r^{t},$
> where we assume that the update direction $\delta$ is constant.
> which yields
> $ h_t = \delta n_0 \frac{1 - r^{t}}{1 - r}, h_t^{(\mathrm{no-decay})} = \delta n_0 t. $
> Thus, their norms satisfy
> $ ||h_t^{(\mathrm{no-decay})}|| =  ||h_t||\frac{(1-r)t}{1 - r^{t}}. $
>
> **Intuitively, the point at which $ ||h_t^{(\mathrm{no-decay})}||$ begins to increase corresponds to the stage where the global gradient no longer decreases, indicating that FedDyn has reached a basic level of convergence.**
>
> | Decay \(r\) | Switch Round   | FedAvg  | FedDyn | FedDyn → FedAvg |
> |-------------|----------------|---------|--------|------------------|
> | 0.997       | 337 / 1000     | (62.58) | 65.43  | 65.62           |
> | 0.998       | 320 / 1000     | (62.58) | 65.68  | 65.40           |
> | 0.999       | 330 / 1000     | (62.58) | 65.37  | 65.18           |
> | 1.000       | 338 / 1000     | (62.58) | 65.58  | 64.31           |
>
> We observe that, although the learning-rate decay varies (e.g.,
> $r\in\{1.0, 0.999, 0.998, 0.997\}$), the resulting effective pre-train
> steps exhibit a consistent switching point. In practice, leveraging only ~**30%** of the
> regularization period is already sufficient to obtain nearly the same
> accuracy as using FedDyn for all communication rounds. While extending
> the duration of regularization can potentially yield even higher
> accuracy than standard FedDyn, this comes at a greater
> computational cost.
> We show that, even in the cross-silo setting, a practical guideline for determining the switching point can be computed, as summarized in the table below.
>
> | Decay \(r\) | Switching Round | FedAvg | FedDyn | FedDyn → FedAvg |
> |-------------|------------------|--------|--------|------------------|
> | 0.99        | 76 / 300         | 66.30  | 66.03  | 68.26           |
>
> Our proposed guideline therefore provides a
> practical and efficient lower bound: a minimal switching schedule that
> achieves (at least) FedDyn-level performance with substantially reduced
> computation.

---

> ### Author Response · Authors · 2025-11-21
> **Response to reviewer PREy (2/2)**
>
> **W3-Q2-Q3. Limited Generality: The conclusion that only FedDyn serves as a good initializer is not fully explained, limiting the generality of the "regularization for initialization" principle to other methods. What is the fundamental reason that FedDyn succeeds as an initializer while a similar method like SCAFFOLD fails? Is the two-stage training principle generalizable to other combinations of FL algorithms beyond FedDyn$\rightarrow$FedAvg?**
>
> As the reviewer correctly pointed out, our analysis focuses primarily on FedDyn among the regularization-based methods.
> Among the methods we tested, FedDyn was also the only approach that demonstrated clear potential as an initialization mechanism for federated learning.
> Our perspective of rethinking regularization from the viewpoint of initialization does not imply that all regularization-based methods should converge to a good initialization point in federated learning.
> Rather, our intended message is that, in principle, an ideal regularization method should exhibit a tendency to converge toward a global stationary point.
> **While many regularization-based approaches attempt to mitigate client drift, only FedDyn explicitly ensures that the local objective functions become non-conflicting in the asymptotic region and guarantees convergence toward the global stationary point.** Our comparative analysis from the perspectives of gradient diversity and feature learning further supports this intuition.
> For example, when training begins from a good initialization point, such as one obtained by pre-training on public data, the resulting gradient diversity remains relatively low[1].
> In Figure 1(b), FedDyn is the only regularization-based method that exhibits this desirable behavior, differentiating it clearly from other approaches.
>
> In summary, we do not claim that all existing regularization-based methods converge to good initialization points in federated learning.
> **However, if an ideal form of regularization were able to perfectly align the local objectives and guide the optimization trajectory toward the true global stationary point, analogous to pre-training on high-quality public data, this would naturally yield a strong initialization.**
> From this viewpoint, FedDyn is unique in that it asymptotically aligns the local and global objectives, and our experiments confirm this through analyses of gradient diversity and feature learning behaviors.
>
> [1] Where to begin? on the impact of pre-training and initialization in federated learning. ICLR 23.
>
> **W1. Lack of Theoretical Proof: The paper lacks a formal convergence proof for the proposed two-stage FedDyn$\rightarrow$FedAvg algorithm, relying instead on empirical results.**
>
> The convergence of algorithms such as FedAvg and FedDyn has already been established in prior work, and our goal is not to provide new convergence proofs or tighter rates.
> Instead, we analyze practical algorithmic behaviors that existing theory does not capture.
> For instance, FedDyn’s formulation assumes that local training reaches a stationary point at every iteration, whereas real FL settings use only a few local steps, creating a gap between theory and practice.
> Our work therefore focuses on examining FedDyn’s regularization mechanism, and comparing it with other regularization-based methods, from perspectives not revealed by standard convergence analyses.

---

### Official Review · Reviewer_c19g · 2025-10-31

**Soundness:** 2
**Presentation:** 3
**Contribution:** 2
**Rating:** 4
**Confidence:** 4

**Summary:**

This paper re-examines the role of regularization in federated learning (FL) from an initialization perspective. Through a comparative analysis of client gradients and learned features, the authors find that FedDyn is the most effective regularization method for mitigating client drift caused by data heterogeneity. However, they argue that all practical regularization methods, including FedDyn, are imperfect approximations of an ideal scheme, leading to side effects and additional overhead that diminish their benefits in later training stages. Based on the observation that FL is less sensitive to heterogeneity when well-initialized, the paper proposes a two-stage training strategy: using FedDyn for pre-training to provide a strong initialization, followed by standard FedAvg for fine-tuning. Experiments in both cross-silo and cross-device settings demonstrate that this approach achieves faster convergence and higher accuracy compared to using regularization throughout the entire training process.

**Strengths:**

* The paper provides a good observational analysis of how different regularization methods impact client gradients and learned features in federated learning.
* The writing is clear and the paper is well-structured.
* The analysis is supported by abundant experimental figures, providing a multi-faceted view of the regularization effects.

**Weaknesses:**

* The paper claims that regularization encourages local models to learn features that better align with the global model, but this claim is not supported by any theoretical convergence analysis.
* The "side effects" of regularization are not clearly explained. The paper asserts that the server control variate in FedDyn "does not accurately approximate the gradient of the global objective function", but the reasoning is not fully developed. Furthermore, the claim of significant "additional computational cost" is not quantified. The overhead of adding a regularization term, which is often just a vector operation, seems marginal compared to the cost of model training.
* The algorithms discussed (FedAvg, FedDyn, etc.) are all well-established. The main contribution appears to be the proposed two-stage training strategy, which is a combination of existing methods. The novelty of this contribution seems limited.
* The paper provides a formal criterion for switching from FedDyn to FedAvg, but its practical application is unclear. The criterion depends on the term Ct, which seems difficult to compute in a real experiment. How is the switching point determined in the experiments in real experiments? The compution of Ct in the experiment is inpractical. A sensitivity analysis on the switching point would be beneficial.

**Questions:**

* In Section 3, why is the analysis based on the "pseudo-gradient" instead of the exact gradient?
* How to understand Figure 7 (c) and (d)? It seems the proposed algorithm is not competitive.

---

> ### Author Response · Authors · 2025-11-21
> **Response to reviewer c19g (1/2)**
>
> We thank the reviewer for the constructive and insightful comments. We address each concern in detail below.
>
> **W 2-1. The side effects of regularization are not clearly explained. The paper asserts that the server control variate in FedDyn does not accurately approximate the gradient of the global objective function, but the reasoning is not fully developed.**
>
> A more detailed explanation of the side effects of FedDyn is as follows.
> We will revise the manuscript to elaborate on these points more clearly in the next version.
> The discrepancy between the server-side control variate $h_t$ and the true global gradient arises because $h_t$ is, by design, only an approximate representation rather than the exact global gradient.
> Client drift, defined as $\nabla L_k(\theta_k^t)$, should ideally correspond to the gradient of each client's objective function.
> However, in practice it is computed through the relation
> $
> \nabla L_k(\theta_k^{t+1}) = \nabla L_k(\theta_k^{t}) - \alpha \left( \theta_k^{t+1} - \theta^{t} \right),
> $
> which is derived under the assumption that local training converges to a stationary point.
> In reality, with only a few local epochs, local training does not converge exactly to such a stationary point, making this approximation inaccurate.
> Since $h_t$ is defined as $\sum_{k} \nabla L_k(\theta_k^t)$, it can only approximate the gradient of the global objective function.
>
> **W 2-2. The claim of significant additional computational cost is not quantified. The overhead of adding a regularization term, which is often just a vector operation, seems marginal compared to the cost of model training.**
>
> Regularization-based methods such as FedDyn and SCAFFOLD add extra terms to the local objective, resulting in noticeably higher computational cost than the simple loss in FedAvg.
> In our setup, FedDyn required about 50\% more computation, and prior work[1] also identifies FedDyn as one of the most computation-heavy FL algorithms.
> The exact overhead depends on hardware and model architecture; for example, on an A100 GPU with ResNet-18, FedDyn introduces over 40\% additional cost.
>
> [1] Not all federated learning algorithms are created equal: A performance evaluation study. arXiv 2024.
>
> **Q1. In Section 3, why is the analysis based on the pseudo-gradient instead of the exact gradient?**
>
> Because clients perform multiple local updates, we use the accumulated update (the pseudo gradient) to represent their overall update direction.
> If clients instead performed only a single step using the exact gradient, data heterogeneity would not cause any issue.
>
> **Q2. How to understand Figure 7 (c) and (d)? It seems the proposed algorithm is not competitive.**
>
> In our experiments, we use ImageNet pre-training as an upper bound and compare it with FedDyn-based initialization.
> While ImageNet pre-training assumes access to a task-relevant public dataset, often unrealistic in privacy-sensitive FL, FedDyn requires no public data and is thus broadly applicable.
> Under IID conditions, FedDyn initialization performs comparably to the upper bound, but in Non-IID settings, ImageNet pre-training clearly outperforms FedDyn, showing that high-quality pre-training becomes increasingly important as data heterogeneity grows.
>
> **W1. The paper claims that regularization encourages local models to learn features that better align with the global model, but this claim is not supported by any theoretical convergence analysis.**
>
> The convergence of algorithms such as FedAvg and FedDyn has already been established in prior work, and our goal is not to provide new convergence proofs or tighter rates.
> Instead, we analyze practical algorithmic behaviors that existing theory does not capture.
> For instance, FedDyn’s formulation assumes that local training reaches a stationary point at every iteration, whereas real FL settings use only a few local steps, creating a gap between theory and practice.
> Our work therefore focuses on examining FedDyn’s regularization mechanism, and comparing it with other regularization-based methods, from perspectives not revealed by standard convergence analyses.
>
> **W3. The algorithms discussed (FedAvg, FedDyn, etc.) are all well-established. The main contribution appears to be the proposed two-stage training strategy, which is a combination of existing methods. The novelty of this contribution seems limited.**
>
> As the reviewer noted, our work does not propose a new algorithm but provides a deeper analysis of existing regularization-based methods and introduces a new perspective on using regularization through initialization.
> We show that applying regularization only in the early phase achieves performance comparable to using it throughout training, while removing it in later stages can even improve results.
> This challenges the common assumption that heavy regularization must be applied uniformly across the entire training process.

---

> ### Author Response · Authors · 2025-11-21
> **Response to reviewer c19g (2/2)**
>
> **W4. The paper provides a formal criterion for switching from FedDyn to FedAvg, but its practical application is unclear. How is the switching point determined in the experiments in real experiments? A sensitivity analysis on the switching point would be beneficial.**
>
> As the reviewer noted, we used a fixed switching point because it already provides strong performance, and thus we did not emphasize the link between the theoretical and practical criteria.
> We will include a concise discussion of this connection in the revised manuscript.
> While Appendix A.2 presents a sensitivity analysis of the switching point, we summarize the key findings here and introduce a practical guideline for selecting the switching point based on our theoretical analysis.
> Assuming that FedDyn incurs 50\% higher computational cost than FedAvg, the results show that delaying the switch improves final accuracy but increases computation and the number of rounds needed to reach the target accuracy.
>
> | Switch round     | 0 (FedAvg) | 1000 (FedDyn) | 150   | 300   | 450   | 600   | 750   | 900   |
> |------------------|------------|---------------|-------|-------|-------|-------|-------|-------|
> | Final accuracy   | 62.58      | 65.71         | 63.87 | 64.98 | 65.94 | 66.99 | 67.13 | 67.71 |
> | Rounds to 67%    | x          | x             | x     | x     | x     | 660   | 790   | 910   |
> | Computation cost | 1          | 1.5           | 1.075 | 1.15  | 1.225 | 1.3   | 1.375 | 1.45  |
>
> However, determining an appropriate pre-training duration in practice is not straightforward.
> Building on the theoretical switching point, we therefore propose a practical guideline for selecting the pre-train step.
> **Although there is an inherent trade-off between computation and performance, our aim is to provide a principled criterion that ensures sufficient performance while avoiding unnecessary regularization.**
>
> Proposition 1 states that the switching point should occur when $C_t$ exceeds its historical average.
> Since $C_t$ cannot be computed directly, we instead use a practical proxy: the server control variate $||h_t||$ should surpass its historical average.
> Because $||h_t||$ starts from zero and fluctuates during the early stage of training, we compute a slope over a 50-round window and identify the first interval in which the slope becomes positive; this interval is designated as the switching point.
> When the learning rate decays by a factor $r$, the magnitude of $h_t$ decreases, making this interval more difficult to detect.
> To compensate for this effect, we approximate the hypothetical trajectory under a constant learning rate.
> Consider the update rule $ h_{t+1} = h_t + n_t \delta,  n_t = n_0 r^{t},$
> where we assume that the update direction $\delta$ is constant.
> which yields
> $ h_t = \delta n_0 \frac{1 - r^{t}}{1 - r}, h_t^{(\mathrm{no-decay})} = \delta n_0 t. $
> Thus, their norms satisfy
> $ ||h_t^{(\mathrm{no-decay})}|| =  ||h_t||\frac{(1-r)t}{1 - r^{t}}. $
>
> **Intuitively, the point at which $ ||h_t^{(\mathrm{no-decay})}||$ begins to increase corresponds to the stage where the global gradient no longer decreases, indicating that FedDyn has reached a basic level of convergence.**
>
> | Decay \(r\) | Switch Round   | FedAvg  | FedDyn | FedDyn → FedAvg |
> |-------------|----------------|---------|--------|------------------|
> | 0.997       | 337 / 1000     | (62.58) | 65.43  | 65.62           |
> | 0.998       | 320 / 1000     | (62.58) | 65.68  | 65.40           |
> | 0.999       | 330 / 1000     | (62.58) | 65.37  | 65.18           |
> | 1.000       | 338 / 1000     | (62.58) | 65.58  | 64.31           |
>
> We observe that, although the learning-rate decay varies (e.g.,
> $r\in\{1.0, 0.999, 0.998, 0.997\}$), the resulting effective pre-train
> steps exhibit a consistent switching point. In practice, leveraging only ~**30%** of the
> regularization period is already sufficient to obtain nearly the same
> accuracy as using FedDyn for all communication rounds. While extending
> the duration of regularization can potentially yield even higher
> accuracy than standard FedDyn, this comes at a greater
> computational cost.
> We show that, even in the cross-silo setting, a practical guideline for determining the switching point can be computed, as summarized in the table below.
>
> | Decay \(r\) | Switching Round | FedAvg | FedDyn | FedDyn → FedAvg |
> |-------------|------------------|--------|--------|------------------|
> | 0.99        | 76 / 300         | 66.30  | 66.03  | 68.26           |
>
> Our proposed guideline therefore provides a
> practical and efficient lower bound: a minimal switching schedule that
> achieves (at least) FedDyn-level performance with substantially reduced
> computation.

---

### Official Review · Reviewer_GRHu · 2025-11-01

**Soundness:** 3
**Presentation:** 3
**Contribution:** 2
**Rating:** 6
**Confidence:** 3

**Summary:**

The paper presents a compelling argument for rethinking the role of regularization in FL. Its core thesis is that while regularization methods like FedDyn are highly effective at mitigating client drift, they are computationally expensive and their benefits diminish after the model is well-initialized. The authors' key proposal—a two-stage training strategy that uses FedDyn only for pre-training before switching to standard FedAvg—is novel, practical, and well-supported by experimental evidence from gradient and feature-learning perspectives.

**Strengths:**

1. The paper goes beyond mere accuracy plots. The analysis from gradient perspective (diversity, cosine similarity) and feature perspective (interaction tensor) provides a much deeper, mechanistic understanding of why FedDyn works better than other methods. This is a major strength.
2. The paper is well-structured and easy to follow.

**Weaknesses:**

1. The paper states that FedDyn has "side effects" (e.g., the server control variate inaccurately approximates the global gradient, negatively impacting features). However, this is not demonstrated as clearly as its benefits.
2. The analysis focuses heavily on FedDyn, with SCAFFOLD, MOON, and FedNTD as comparisons. While justified by FedDyn's performance, a broader discussion of why this two-stage strategy might or might not work for other state-of-the-art methods (e.g., FedProx) would strengthen the generalizability of the claim.

**Questions:**

Please refer to weaknesses for details.

---

> ### Author Response · Authors · 2025-11-21
> **Response to reviewer GRHu**
>
> We thank the reviewer for the constructive and insightful comments. We address each concern in detail below.
>
> **W1. The paper states that FedDyn has "side effects" However, this is not demonstrated as clearly as its benefits.**
>
> A more detailed explanation of the side effects of FedDyn is as follows.
> We will revise the manuscript to elaborate on these points more clearly in the revised version.
> First, the server-side control variate $h_t$ does not accurately reflect the true global gradient.
> Second, this inaccurate representation of the global gradient by $h_t$ directly leads to the observed side effects.
> The discrepancy between the server-side control variate $h_t$ and the true global gradient arises because $h_t$ is, by design, only an approximate representation rather than the exact global gradient.
> Client drift, defined as $\nabla L_k(\theta_k^t)$, should ideally correspond to the gradient of each client's objective function.
> However, in practice it is computed through the relation
> $
> \nabla L_k(\theta_k^{t+1}) = \nabla L_k(\theta_k^{t}) - \alpha \left( \theta_k^{t+1} - \theta^{t} \right),
> $
> which is derived under the assumption that local training converges to a stationary point.
> In reality, with only a few local epochs, local training does not converge exactly to such a stationary point, making this approximation inaccurate.
> Since $h_t$ is defined as $\sum_{k} \nabla L_k(\theta_k^t)$, it can only approximate the gradient of the global objective function.
>
> The issue that arises when $h_t$ does not accurately reflect the global gradient can be understood both theoretically and empirically.
> Theoretically, the expected update of the server-side variable $\gamma^t = \sum_k \theta_k^t$ in one round is given by
> $
> E\left[\gamma_t - \gamma_{t-1}\right] = \frac{1}{\alpha m} \sum_{k \in [m]} E\left[ - \nabla L_k(\tilde{\theta}_k^t) \right]$
> which directly depends on $h_t$.
> In practice, FedDyn updates the server model by moving in the direction of $-\frac{1}{\alpha} h_t$, making the algorithm highly sensitive to the accuracy of $h_t$ as a proxy for the true global gradient.
> As illustrated in Figure8 of the Appendix, this inaccuracy can cause the global model $\theta^t$ to be updated in a direction that is misaligned with the ideal $\gamma^t$ trajectory, ultimately leading to inferior performance.
> This empirical evidence further supports that the deviation of $h_t$ from the true global gradient results in undesirable side effects during training.
>
> **W2. The analysis focuses heavily on FedDyn, with SCAFFOLD, MOON, and FedNTD as comparisons. While justified by FedDyn's performance, a broader discussion of why this two-stage strategy might or might not work for other state-of-the-art methods (e.g., FedProx) would strengthen the generalizability of the claim.**
>
> As the reviewer correctly pointed out, our analysis focuses primarily on FedDyn among the regularization-based methods.
> Among the methods we tested, FedDyn was also the only approach that demonstrated clear potential as an initialization mechanism for federated learning.
> Our perspective of rethinking regularization from the viewpoint of initialization does not imply that all regularization-based methods should converge to a good initialization point in federated learning.
> Rather, our intended message is that, in principle, an ideal regularization method should exhibit a tendency to converge toward a global stationary point.
> **While many regularization-based approaches attempt to mitigate client drift, only FedDyn explicitly ensures that the local objective functions become non-conflicting in the asymptotic region and guarantees convergence toward the global stationary point.** Our comparative analysis from the perspectives of gradient diversity and feature learning further supports this intuition.
> For example, when training begins from a good initialization point, such as one obtained by pre-training on public data, the resulting gradient diversity remains relatively low[1].
> In Figure 1(b), FedDyn is the only regularization-based method that exhibits this desirable behavior, differentiating it clearly from other approaches.
>
> In summary, we do not claim that all existing regularization-based methods converge to good initialization points in federated learning.
> **However, if an ideal form of regularization were able to perfectly align the local objectives and guide the optimization trajectory toward the true global stationary point, analogous to pre-training on high-quality public data, this would naturally yield a strong initialization.**
> From this viewpoint, FedDyn is unique in that it asymptotically aligns the local and global objectives, and our experiments confirm this through analyses of gradient diversity and feature learning behaviors.
>
> [1] Where to begin? on the impact of pre-training and initialization in federated learning. ICLR 23.

---

### Author Response · Authors · 2025-12-02
**Summary of Core Contributions and Responses to Reviewers**

We thank all reviewers for their insightful and constructive feedback.
Our rebuttal clarified the motivation, empirical validation, and broader significance of the proposed two-stage training strategy (FedDyn → FedAvg),
which reinterprets regularization as a means of obtaining an effective initialization rather than a rule that must persist throughout training.
This summary presents our core idea together with concise responses to the key issues raised by the reviewers.

**1. Core Idea**

Regularization-based methods (FedDyn, SCAFFOLD, MOON, FedNTD) aim to reduce client drift,
but only FedDyn aligns local and global objectives asymptotically.
This makes FedDyn function as a data-free pre-training stage that moves the global model near a global stationary point.
Once this is achieved, switching to a simpler optimizer such as FedAvg maintains or even improves accuracy with much lower cost.

**2. Limited Generality**

A key contribution of this work is a comparative analysis of regularization methods beyond accuracy,
considering gradient diversity, cosine similarity, and feature learning.
Although all methods target data heterogeneity, they exhibit distinct training dynamics.
FedDyn uniquely shows lower gradient diversity and better feature alignment, confirming that its regularization implicitly guides models toward globally consistent representations.
Among the methods we tested, FedDyn was also the only approach that demonstrated clear potential as an initialization mechanism for federated learning.
**Our perspective of rethinking regularization from the viewpoint of initialization does not imply that all regularization-based methods should converge to a good initialization point in federated learning.**
Rather, our intended message is that, in principle, an ideal regularization method should exhibit a tendency to converge toward a global stationary point.
While many regularization-based approaches attempt to mitigate client drift, only FedDyn explicitly ensures that the local objective functions become non-conflicting in the asymptotic region and guarantees convergence toward the global stationary point.
Our comparative analysis from the perspectives of gradient diversity and feature learning further supports this intuition.

**3. Side Effects and Cost**

The server control variate $h_t$ in FedDyn only approximates the global gradient;
incomplete local convergence can cause misaligned updates, explaining minor side effects.
There will inevitably be differences between an ideal regularization that perfectly eliminates client drift and the practical solutions that approximate it under real-world constraints.
It also introduces about 40–50% more computation than FedAvg, motivating its use mainly as an early-phase initializer.

**4. Switching Criterion**

In the main text, we introduced the switching point at which the gain from regularization is offset by the cost induced by inaccurate updates of $h_t$.
Several reviewers asked for a more practical guideline on how this switching point can be determined in real-world scenarios.
In practice, there is no perfect switching point, as it inherently involves a trade-off between final accuracy and computational efficiency.
Nevertheless, a practical rule can be applied, which aligns with the switching criterion described above:
transition when $||h_t||$ stops decreasing, indicating that the approximated global gradient norm has stabilized.
In practice, leveraging only ~30% of the regularization period is already sufficient to obtain nearly the same accuracy as using regularization for all communication rounds, providing a clear and reproducible guideline for determining the switching point.

**5. Comparisons and Scope**

Experiments with FedAlign, FedDecorr, and FedFA show that FedDyn achieves higher accuracy,
and the two-stage FedDyn → FedAvg strategy consistently outperforms continuous regularization.
Our goal is analysis, not a new convergence proof; we instead bridge the gap between theoretical assumptions and realistic federated settings.

**6. Broader Implication**

This study challenges the notion that strong regularization must persist throughout training.
We show that early-phase regularization is sufficient to stabilize learning,
while switching to FedAvg later enhances efficiency and final accuracy.

---

### Meta-Review · Area_Chair_LXDt · 2025-12-26

**Summary:**

The submission advances a simple thesis: heavy client regularization in federated learning is most valuable early, and should be used mainly to obtain a better starting point for subsequent training. Based on this view, the paper recommends a two-stage schedule in which FedDyn is used for an initial phase and then training switches to FedAvg to reduce overhead while preserving (or sometimes improving) final accuracy.

Reviewers appreciated the effort to look beyond final accuracy and analyze training dynamics using gradient and feature-based diagnostics. Several saw the two-stage strategy as a practical approach. At the same time, they raised recurring concerns about limited novelty, unclear practical guidance for the switching point, limited generality beyond FedDyn, and incomplete support for claims about side effects and cost.

Overall, I do not recommend acceptance. Even after the rebuttal clarifications and added experiments, the empirical evidence and experimental protocol are not yet strong enough to support the paper’s broader claims. The main “best regularizer” claim rests on bespoke proxy metrics whose meaning and robustness are not established, and the evaluation protocol is not consistently controlled across settings. The empirical comparisons also lack a clearly described tuning protocol across methods, which is particularly concerning given the paper’s own evidence that FedDyn is sensitive to its alpha and that hyperparameter search is needed. Finally, the competitor set is explicitly restricted to a small subset of methods, and the paper adopts FedDyn as the default early, which makes the overall positioning feel too strong relative to the breadth of the federated learning literature.

**Reviewer Concerns:**

The rebuttal and discussion address several concrete points. The authors clarify the intended scope, add sensitivity results for the switching point, and provide a more detailed explanation of why the server control variate can be an imperfect proxy for the global gradient, motivating a switch to FedAvg. They also add some additional switching combinations and a limited ablation of second-stage choices.

However, the decision driving concerns remain outstanding. The core evaluation metrics are not validated as reliable benchmark criteria, and their specific measurement choices make it difficult to interpret them as neutral evidence rather than supportive proxies. The experimental methodology does not describe how baselines were tuned or how method-specific hyperparameters were selected, which undermines both fairness and reproducibility.

In addition, the switching point is a major driver of both final accuracy and cost, so the proposal introduces a critical hyperparameter whose optimal value is likely task-dependent and method pair-dependent. To resolve the remaining concerns, the paper would need a clearly stated tuning protocol for all compared methods, matched compute and communication reporting, and stronger robustness evidence showing that the conclusions hold under reasonable retuning and across a broader baseline set.

**Reviewer Scores:**

c19g and NfTo are the most likely to increase. The rebuttal provides more detail on the switching choice and includes additional switching combinations, which address their main requests.

GRHu is already positive about the paper, and might not increase further.

PREy is unlikely to change: their review questioned the overall research contribution and generality, and the rebuttal does not fundamentally change that picture.

---

### Decision · Program_Chairs · 2026-01-26

Reject